# Efficient Heuristics Generation for Solving Combinatorial Optimization Problems Using Large Language Models

## Abstract

Recent studies exploited Large Language Models (LLMs) to autonomously generate heuristics for solving Combinatorial Optimization Problems (COPs), by prompting LLMs to first provide search directions and then derive heuristics accordingly. However, the absence of task-specific knowledge in prompts often leads LLMs to provide unspecific search directions, obstructing the derivation of well-performing heuristics. Moreover, evaluating the derived heuristics remains resource-intensive, especially for those semantically equivalent ones, often requiring unnecessary resource expenditure. To enable LLMs to provide specific search directions, we propose the Hercules algorithm, which leverages our designed Core Abstraction Prompting (CAP) method to abstract the core components from elite heuristics and incorporate them as prior knowledge in prompts. We theoretically prove the effectiveness of CAP in reducing unspecificity and provide empirical results in this work. To reduce the required computing resources for evaluating the derived heuristics, we propose few-shot Performance Prediction Prompting (PPP), a first-of-its-kind method for the Heuristic Generation (HG) task. PPP leverages LLMs to predict the fitness values of newly derived heuristics by analyzing their semantic similarity to previously evaluated ones. We further develop two tailored mechanisms for PPP to enhance predictive accuracy and determine unreliable predictions, respectively. The use of PPP makes Hercules more resource-efficient and we name this variant Hercules-P. Extensive experiments across various HG tasks, COPs, and LLMs demonstrate that Hercules outperforms the state-of-the-art LLM-based HG algorithms, while Hercules-P excels at minimizing computing resources. In addition, we illustrate the effectiveness of CAP, PPP, and the other proposed mechanisms by conducting relevant ablation studies.

## 1 Introduction

Heuristic algorithms have long been a preferred approach for solving Combinatorial Optimization Problems (COPs) (Rego et al., 2011). To automate the derivation of heuristics for a given COP, Heuristic Generation (HG) methods have attracted significant attention (Burke et al., 2013). Early HG methods predominantly employ Evolutionary Computation (EC) algorithms to derive heuristics. However, these methods focus on the exploration and exploitation in the micro search space composed of the predefined modules, often resulting in limited performance (Ye et al., 2024a).

Recently, the emergence of Large Language Models (LLMs) has facilitated the autonomous derivation of heuristics, eliminating the need for manually defining the search space (Liu et al., 2023a; 2024a; van Stein & Bäck, 2024). In addition, compared to conventional EC algorithms, LLMs benefit from a broader search space by leveraging their mega-size training corpora, resulting in elevated performance (Yang et al., 2024; Ma et al., 2024; Liu et al., 2024b). Specifically, these LLM-based HG methods exploit LLMs to provide search directions, which are then used to derive (novel) offspring heuristics (Romera-Paredes et al., 2024). These produced heuristics are subsequently evaluated using COP instances to determine their fitness values, with the better-performing heuristics carried over to the next iteration. For example, Liu et al. (2023a) proposed prompting methods that emulate crossover and mutation operators as search strategies, thereby implicitly providing search directions. To let LLMs offer more explicit search directions, Ye et al. (2024a) proposed Reflection

Figure 1: Illustration of the search directions produced using RP and CAP for the task described in Section 4.1. When RP prompts LLMs (GPT-4o-mini used in this example) for search directions directly, the LLMs may respond with unspecific directions (highlighted in blue). Different from RP, our CAP enhances the quality of the produced search directions by first prompting the LLMs to abstract the core components as prior knowledge in a zero-shot manner (highlighted in red).

Figure 2: Illustration of two heuristics with identical semantics, produced by LLMs (GPT-3.5-turbo used in this example) for the task described in Section 4.2. Code snippets with literal equivalence are highlighted in blue, while those with semantic equivalence are highlighted in pink.

Prompting (RP), which requires LLMs to reflect on the relative performance of the produced heuristics and provide insights as search directions. These directions are then used to derive heuristics with expected elevated performance in subsequent crossover and mutation promptings.

These existing LLM-based HG methods face two key challenges. Firstly, when prompting LLMs to provide search directions (e.g., reflections on the relative performance of heuristics), the lack of task-specific knowledge in prompts often leads to over-generalized, unspecific directions that hinder the derivation of high-performance heuristics. As illustrated in Figure 1(a), the produced search directions *"Understand problem specifics"* and *"test and iterate"* are vague, over-general, and lack actionable steps required for heuristic generation. Consequently, they contribute little to the derivation of high-performance heuristics. In contrast, other elements of the produced search directions are more specific. For example, *"normalize heuristic values"* provides an actionable step that can be directly applied to derive heuristics. Therefore, it is essential to reduce unspecificity in the produced search directions. Secondly, during the search process, LLM-based HG methods often derive numerous heuristics, some of which may be semantically or even literally identical, as illustrated in Figure 2. Reevaluating these heuristics using COP instances (i.e., conventional fitness evaluation method) not only wastes computing resources but also significantly prolongs the search process (Chen et al., 2024). In particular, these heuristics often involve numerous linear operations and conditional branches, which GPUs cannot efficiently accelerate (Wachowiak et al., 2017). In addition, providing LLMs with all historical heuristics to avoid deriving semantically similar ones is impractical. This approach may compel LLMs to derive overly random or unviable heuristics, while significantly increasing the cost of context tokens.

To better address the first challenge, we propose **Heur**istic Generation Using Large Language Model**s** (**Hercules**), which exploits our proprietary, straightforward yet effective Core Abstraction Prompting (**CAP**) method to reduce unspecificity in the produced search directions and thus enable the derivation of high-performance heuristics. Specifically, CAP directs an LLM to abstract the core components from the top-$k$ heuristics (i,e., elite heuristics) in the current population and then provide more specific search directions based on these components (see Section 3.1). Notably, as illustrated in Figure 1(b), CAP operates in a zero-shot manner, abstracting the core components

without providing any examples to guide this abstraction process, which leads to significant savings in context token costs. To couple with CAP, we introduce a rank-based selection mechanism that increases the likelihood of selecting high-performance heuristics as parents (used in the following crossover and mutation promptings), rather than relying on random selection (Ye et al., 2024a). Meanwhile, by incorporating the concept of information gain, we theoretically prove that CAP can reduce unspecificity in the produced search directions in Appendix A.

To better address the second challenge, we propose **Hercules-P**, which integrates CAP with our novel Performance Prediction Prompting (**PPP**) method. PPP operates in a few-shot manner by presenting LLMs with a small set of previously evaluated heuristics as examples and prompting LLMs to predict the fitness values of the newly produced heuristics based on their semantic similarity to the presented examples (see Section 3.2). Therefore, PPP reduces the number of heuristics that require evaluation using COP instances. Generally speaking, to enhance the predictive accuracy of PPP, we can either increase the number of examples or enhance their quality. However, collecting numerous heuristic examples along with their corresponding performance is resource-intensive. This contradicts to the primary purpose of incorporating PPP, which is to reduce resource expenditure during the search process. Moreover, unlike Neural Architecture Search (NAS), which benefits from extensive benchmarks (Ying et al., 2019; Qiu et al., 2023), the HG task lacks benchmarks with pre-evaluated heuristics. Therefore, we opt to provide higher-quality examples through a tailored example selection mechanism, termed EXEMPLAR, which favors distinct parent heuristics with superior performance as examples. Meanwhile, to determine unreliable predictions, we develop the Confidence Stratification (ConS) mechanism that requires the LLM to provide confidence levels for the predicted fitness values, thereby facilitating the identification of heuristics that need reevaluation. In summary, PPP reduces the resource expenditure in heuristic evaluations while maintaining population diversity, making it effective for tasks with a border search space. To the best of our knowledge, **our work proposes the first LLM-based performance predictor for the HG task.**

To assess the performance of the proposed Hercules and Hercules-P algorithms, we conduct extensive experiments on four HG tasks (see Section 4). The experimental results demonstrate that Hercules outperforms the state-of-the-art (SOTA) LLM-based HG algorithms across diverse HG tasks, COPs, and LLMs, without significantly increasing context or generation token costs. By incorporating PPP, Hercules-P significantly reduces the overall search time by 7%∼59% when compared to Hercules, while achieving on-par performance on the gain metric. Finally, ablation studies validate the effectiveness of the proposed rank-based selection mechanism, EXEMPLAR, and ConS.

The key contributions of this work are as follows.

**i)** We propose the zero-shot CAP method, which reduces unspecificity in the LLM-produced search directions, enabling the derivation of high-performance heuristics. We also provide the theoretical proof of CAP's effectiveness in reducing unspecificity by utilizing the concept of information gain.

**ii)** We propose the few-shot PPP method, a first-of-its-kind LLM-based performance predictor specifically designed for HG tasks. PPP predicts the performance of newly produced heuristics by analyzing their semantic similarity to previously evaluated ones. Moreover, we develop two novel mechanisms: EXEMPLAR and ConS, which significantly enhance the overall performance of PPP.

**iii)** The experimental results demonstrate that our proposed Hercules achieves SOTA performance across diverse HG tasks, COPs, and LLMs, while Hercules-P excels at reducing resource expenditure. Finally, ablation study results validate the effectiveness of all proposed methods.

## 2 RELATED WORK

In this section, we review the relevant literature.

### 2.1 LLM-BASED HEURISTIC GENERATION ALGORITHMS

Conventional EC-based HG algorithms search for the optimal combination of the predefined heuristic modules (Keller & Poli, 2007), which often limits their performance. In contrast, LLM-based HG algorithms eliminate the need for predefining the search space, liberating researchers from manual customization and enabling the derivation of high-performance heuristics (Zhang et al., 2024; Wu et al., 2024a; Huang et al., 2024). Specifically, these algorithms begin with a seed heuristic to prompt LLMs to derive multiple heuristics as the initial population (Liu et al., 2023a; 2024a; Ye et al., 2024a). Each heuristic is then evaluated using a set of COP instances, with its performance

serving as its fitness value. During the iterative process, certain heuristics are selected as parents and presented to LLMs to derive (novel) offspring heuristics. This approach emulates the concepts of crossover and mutation, while implicitly providing search directions for the LLMs to derive heuristics. In addition, certain studies exploit LLMs to provide explicit search directions for deriving well-performing heuristics (Ye et al., 2024a). However, these LLM-based HG algorithms overlook the issue of unspecificity in LLM responses (see Figure 1(a)), which can lead to unspecific search directions that do not contribute to discovering high-performance heuristics.

Similar challenges are observed in tasks such as arithmetic and symbolic reasoning, making it crucial to evoke LLM reasoning through a multi-step process and incorporate task-specific knowledge (Yu et al., 2024; Jiang et al., 2024; Lv et al., 2024). For example, Wei et al. (2022) proposed Chain-of-Thought (CoT) prompting, which directs LLMs to emulate the given examples in completing a multi-step solution process, leading to more accurate answers. Subsequently, Zheng et al. (2024) proposed the few-shot Step-back Prompting (SP), which exploits the given examples to enable LLMs to abstract high-level principles and then apply these principles in reasoning. In a similar multi-step fashion, we propose CAP to mitigate unspecificity in the produced search directions for better solving HG tasks. However, unlike CoT and SP, CAP operates in a zero-shot manner, because it abstracts the core components without any examples to guide the abstraction process.

## 2.2 LLM-based Performance Prediction Methods

In the field of NAS, performance predictors, typically Deep Neural Networks, are widely used to reduce search costs by predicting the performance of candidate architectures (Baker et al., 2017; Wu et al., 2021). These predictors model neural architectures as graphs, where nodes represent subnets and edges represent the connections between subnets (Chu et al., 2023; Liu et al., 2022). The graphs are then encoded into vectors, and the mapping between these vectors and the corresponding performance metrics is learned. Recently, Jawahar et al. (2024) and Chen et al. (2024) proposed LLM-based predictors for predicting the performance of neural architectures. Specifically, they employed examples of architectures and corresponding performance metrics to prompt LLMs, leveraging semantic similarity to predict the performance of newly searched architectures.

In the context of HG, conventional performance predictors may struggle to accurately evaluate heuristics due to the difficulty in modeling these diverse and complex heuristics as graph structures. However, the LLM-based predictor presents a promising alternative by eliminating the need for explicit heuristic modeling. Consequently, this paper leverages LLMs to predict the performance of heuristics for effectively solving HG tasks. However, unlike (Jawahar et al., 2024) and (Chen et al., 2024), which relied on a larger number of examples, our PPP emphasizes the use of only the higher-quality examples to improve predictive performance (see Section 3.2 for more details).

## 2.3 Neural Combinatorial Optimization Solvers

Neural Combinatorial Optimization (NCO) refers to a class of Neural Network solvers that either independently solve COPs or collaborate with heuristic algorithms (Bengio et al., 2021; Wu et al., 2022; 2024b; Bogyrbayeva et al., 2024). To enable the derivation of insights from historical COP instances and efficiently handle batches of instances in parallel, researchers have recently developed numerous NCO solvers (Kwon et al., 2020; Lu et al., 2020; Hudson et al., 2022; Chen et al., 2023; Kim et al., 2024; Dernedde et al., 2024). However, these NCO solvers still face several challenges. Two of the most prominent ones are how to improve their generalization capabilities (Zhou et al., 2023; Xiao et al., 2024; Hottung et al., 2024) and their performance on large-scale COPs (Hou et al., 2023; Sun & Yang, 2023; Min et al., 2023; Ye et al., 2024b). Recently, Wang et al. (2024) proposed a distance-aware heuristic algorithm designed to enhance the generalization ability of NCO solvers trained on small-scale COPs for solving large-scale COPs. To assess the effectiveness of the proposed Hercules and Hercules-P algorithms, we apply them to improve the performance of two classic NCO solvers on both small-scale and large-scale COPs in Section 4.4.

## 3 Heuristic Generation with Hercules and Hercules-P

The illustrations of Hercules and Hercules-P are schematically presented in Figure 3. In this section, we first introduce CAP, which is designed to provide more specific search directions for deriving heuristics. We then prove that CAP can reduce unspecificity of the produced search directions. Finally, we present the design of PPP, along with tailored EXEMPLAR and ConS mechanisms.

Figure 3: Overview of the proposed Hercules and Hercules-P algorithms. Hercules exploits CAP to provide specific search directions, which are then used to guide LLMs in deriving high-performance heuristics. In Hercules, the performance of all derived heuristics on a set of COP instances determines their respective fitness values. In contrast, Hercules-P evaluates only a subset of the produced heuristics with COP instances, while the rest are assessed using the proposed PPP method.

## 3.1 CORE ABSTRACTION PROMPTING (CAP)

As aforementioned, when LLMs are tasked with providing search directions, they often generate directions that lack specificity for heuristic derivation. As illustrated in the RP example in Figure 1(a), certain directions, such as *"Understand problem specifics"* and *"test and iterate"*, lack relevance to heuristic derivation and fail to derive well-performing heuristics.

In this case and many others, providing prior knowledge in prompts can help LLMs reduce unspecificity in their responses, leading to more focused, specific search directions. To achieve this, we propose the zero-shot CAP method, which can abstract the core components from the top-$k$ heuristics in the current population without additional guidance. Because the core components are essential for heuristic performance (Xue et al., 2016; Liu et al., 2024a), leveraging them enables LLMs to provide more specific search directions. As shown in Figure 1(b), the suggested direction *"Normalize penalties relative to overall distance"* may lead to more effective heuristic generation (see Appendix B for more comparative examples of search directions produced by RP and CAP). In addition, CAP abstracts the core components once per iteration, instead of abstracting distinct components separately for crossover and elitist mutation operators. Consequently, this approach helps prevent a significant increase in context and generation token costs compared to RP (see Table 2). The details about the adopted crossover and elitist mutation operators, along with other EC definitions, are presented in Appendix C.

In the field of information theory, the advantage of CAP can be quantified using the concept of information gain. In the prior study (Hu et al., 2024), information gain was defined as the reduction in entropy between two states. Extending this concept, we use information gain to quantify entropy reduction in scenarios with and without abstraction, facilitating the assessment of CAP in reducing unspecificity. Specifically, the entropy without abstraction (i.e., the core components are not presented to LLMs) in the $t$th iteration is defined as follows:

$$H(\Omega_t) = -\sum\nolimits_{i:\omega_i \in \Omega_t} p(\omega_i|\Omega_t) \log p(\omega_i|\Omega_t), \tag{1}$$

where $\omega_i$ denotes a direction belonging to the set of all possible directions $\Omega_t$.

When the core components are used as prior knowledge in prompts, an LLM can provide more specific, subdivided search directions either based on one of these core components or disregarding all core components. Consequently, the set of all possible directions, $\Omega_t$, can be partitioned into mutually exclusive subsets, $\Omega_j$, where $\bigcup_{j=0}^{k} \Omega_j = \Omega_t$. Here, when $j \in \{0, 1, \ldots, k-1\}$, $\Omega_j$ represents the subset of directions associated with the $j$th core component (for simplicity, we assume a one-to-one correspondence between core components and heuristics), while $j = k$ corresponds to the subset of directions independent of any core component.

Assuming that the produced direction belongs to the $j$th subset ($j \in \{0, 1, \ldots, k\}$) after providing the core components, the remaining entropy is defined as follows:

$$H(\Omega_j) = -\sum\nolimits_{i:\omega_i \in \Omega_j} p(\omega_i|\Omega_j) \log p(\omega_i|\Omega_j). \tag{2}$$

Then, the entropy with abstraction (i.e., the expected remaining entropy) is defined as $\sum_{j=0}^{k} p_j H(\Omega_j)$, where $p_j$ denotes the probability that the search direction belongs to the $j$th subset, i.e., $p_j = p(\Omega_j)/p(\Omega_t)$. Thus, the information gain from abstracting the core components in the $t$th iteration (the entropy reduction without and with abstraction) is defined as follows:

$$IG(\Omega_t) = H(\Omega_t) - \sum\nolimits_{j=0}^{k} p_j H(\Omega_j). \tag{3}$$

**PPP**

```
Here are some example codes and their corresponding performance scores that you can refer to for prediction: [example_A,
score_A],. . ., [example_B, score_B]. Here is a code that you need to predict: [code].
Predict the performance of the given code by comparing its semantic meaning with the provided example codes. In addition, provide
a confidence level for this code, indicating the degree of semantic similarity to the most relevant example code. The performance
score should be a float within the range [score_A, score_B], the confidence number should be a float within the range [0,1].

[score=10.75, confidence=0.8]
```

Figure 4: Illustration of the prediction process using the proposed PPP method. By analyzing the semantic similarity between the heuristics to be predicted and the previously evaluated ones, LLMs can respond with a performance score for each heuristic with an associated confidence level.

As proven in Appendix A, (3) simplifies to the following expression, ranging from $(0, \log(k+1)]$:

$$IG(\Omega_t) = -\sum\nolimits_{j=0}^{k} p_j \log p_j. \tag{4}$$

Therefore, in theory, providing the core components as prior knowledge in prompts can reduce unspecificity in LLM responses and yield more specific search directions, subsequently leading to heuristics with higher performance.

To fine-search the space with high-quality heuristics, we adopt a rank-based selection mechanism. Specifically, the probability of selecting the $i$th heuristic as a parent is computed as follows:

$$p(x_i) = \frac{1}{\text{rank}(x_i) + N} \bigg/ \sum\nolimits_{j=1}^{N} \frac{1}{\text{rank}(x_j) + N}, \tag{5}$$

where $N$ denotes the population size, and rank$(\cdot)$ returns the rank of the associated fitness value in the ascending order. In addition, Hercules adopts the core components of the top-$k$ heuristics as prior knowledge during the first $\lambda$ percent of iterations ($\lambda \in [0,1]$). In the later iterations, following (Zhan et al., 2009; Yang et al., 2018; Zhang et al., 2021; 2015), to better preserve population diversity, Hercules directly applies the core components of the parent heuristics as prior knowledge to provide search directions, bypassing the abstraction process of elite heuristics.

## 3.2 PERFORMANCE PREDICTION PROMPTING (PPP)

Semantic features have demonstrated significant merits in software engineering tasks, e.g., identifying the defective code regions (Liu et al., 2023b), due to their influence on the overall code performance. Motivated by this concept, we propose the few-shot PPP method, which leverages LLMs to predict the performance of newly produced heuristics by analyzing their semantic similarity to previously evaluated ones, as shown in Figure 4. To achieve higher predictive accuracy with a small number of $N_e$ examples, we propose an example selection mechanism called EXEMPLAR, which operates on a principle similar to providing a more relevant, well-defined knowledge base in retrieval-augmented generation (Gao et al., 2023). Specifically, EXEMPLAR selects the historically best and worst heuristics, i.e., $x_{lb}$ and $x_{ub}$, respectively, as prediction boundaries (assuming the goal of the HG task is to derive the heuristic with the minimum fitness value), while prioritizing parent heuristics with better performance (i.e., lower fitness value). Parent heuristics with better performance are typically more complex and richer in semantic features than those with inferior performance, highly likely leading to higher prediction accuracy. In addition, any heuristic with the same fitness value as a previously selected example will not be chosen as an example. Because if LLMs encounter multiple examples sharing the same fitness value, their predictions may become biased towards this common fitness value, potentially overlooking semantic features. If each example has a distinct fitness value, LLMs can more effectively leverage semantic features to predict the performance of the new heuristics. The set of examples $\mathcal{P}_e$ is selected as follows:

$$\mathcal{P}_e = \{x_{lb}, x_{ub} \mid x_{lb} = \arg\min_{x \in \mathcal{P}_h} f(x), x_{ub} = \arg\max_{x \in \mathcal{P}_h} f(x)\} \cup \{x \mid \arg\text{top}(N_e\text{-}2) \, f(x)\},$$
$$\mathcal{P}_t = \{x \in \mathcal{P}_p \setminus \{x_{lb}, x_{ub}\} \mid f(x_i) \neq f(x_j), \forall i \neq j\}, \tag{6}$$

where $\mathcal{P}_h$ and $\mathcal{P}_p$ denote the set of all historical heuristics and the set of parent heuristics selected from the current iteration according to (5) to produce offspring, respectively, and $f(\cdot)$ denotes the fitness evaluation function, introduced in the following paragraph. EXEMPLAR selects the set $\mathcal{P}_e$ for each iteration.

Nevertheless, LLMs cannot always accurately predict the performance of each heuristic. To mitigate the potential impact of incorrect predictions, we propose the Confidence Stratification (ConS)

---

**Algorithm 1** Hercules-P for Deriving Heuristics

---

**Input:** Maximum iteration number $T$
**Output:** Best heuristic $x_{best}$

1   *//Omitting Steps 5, 10, and 11 makes Hercules-P fall back to the original Hercules algorithm*
2   Initialize and evaluate population $\mathcal{P}$; the number of current iteration $t = 0$
3   **while** $t < T$ **do**
4      Select parent heuristics set $\mathcal{P}_p$ according to (5)    *//Rank-based selection*
5      Select heuristic examples set $\mathcal{P}_e$ for PPP according to (6)    *//EXEMPLAR*
6      **if** $t \le \lambda \cdot T$ **then** Provide search directions using core components of elite heuristics    *//CAP* ;
7      **else** Provide search directions using core components of parent heuristics;
8      Derive heuristics using crossover based on the produced search directions
9      Derive heuristics using elitist mutation based on the produced search directions
10     Predict the fitness values of newly produced heuristics    *//PPP*
11     Determine fitness values $f(\cdot)$ according to (7)    *//ConS*
12     Update $\mathcal{P}$ and $x_{best}$ with new heuristics

---

mechanism. Other than the LLM-predicted fitness value $\xi_i$, ConS prompts an LLM to provide a corresponding confidence level $\phi_i \in [0,1]$ based on the degree of semantic similarity between $x_i$ and the most similar examples in $P_e$. Subsequently, based on $\phi_i$, ConS selectively accepts the predicted fitness values of certain heuristics, while others are reevaluated using COP instances. Intuitively, we implement the following design. For heuristic $x_i$, if $\phi_i$ is sufficiently high, ConS deems $\xi_i$ accurate. If $\phi_i$ is moderately high, only the top-ranked candidates in this category should be trusted to directly adopt $\xi_i$ without reevaluation, reflecting the degraded confidence level. For low $\phi_i$ values, they can only be directly adopted if $\xi_i$ is greater than a predetermined threshold. Because for these heuristics with an acceptable yet sub-par performance score and a not-too-low confidence level, it is intuitive to deem them having inferior performance, without the need for precise predictions (Xu et al., 2021). Specifically, we heuristically define this threshold gauging the known prediction boundaries, i.e., $lb_t$ and $ub_t$. When $\phi_i$ is extremely low, $\xi_i$ is deemed unreliable and the corresponding heuristic must be reevaluated. Such design is implemented as follows to define the fitness function $f(x_i)$:

$$f(x_i) = \begin{cases} \xi_i, & \phi_i \ge 1 - \delta, \\ \xi_i, & 1 - 2\delta \le \phi_i < 1 - \delta \ \wedge \ x_i \in \underset{x \in \mathcal{P}_c}{\arg \text{top}(m_t)}\, \phi(x), \\ \xi_i, & 1 - 3\delta \le \phi_i < 1 - 2\delta \ \wedge \ \xi_i > lb_t + 3\delta(ub_t - lb_t), \\ \mathcal{F}(x_i), & \text{otherwise}, \end{cases} \quad (7)$$

where $\delta \in [0, 1/3]$ denotes a predefined interval to distinguish the performance range of the produced heuristics (a smaller $\delta$ value means ConS only accepts the predicted scores with the highest confidence), $\mathcal{P}_c$ denotes the set of heuristics whose $\phi_i$ values lie within the $[1 - 2\delta, 1 - \delta)$ interval, and $\mathcal{F}(\cdot)$ denotes the conventional fitness evaluation function, which uses COP instances to evaluate heuristics. Furthermore, we gradually decrease the number of heuristics that do not require reevaluation in $\mathcal{P}_c$ after each iteration. Specifically, we set an acceptance threshold $m_t = \lfloor \alpha \cdot \beta^t \cdot N_o \rfloor$, where $\alpha, \beta \in (0, 1)$, and $N_o$ denotes the number of the produced heuristics in the current iteration.

The pseudocode of Hercules-P is presented in Algorithm 1, and its source code is available online[1].

## 4   EXPERIMENTAL RESULTS

This section presents extensive experimental results on various HG tasks, COPs, and LLMs to assess the performance of both Hercules and Hercules-P. Please refer to Appendices D, E, F, and G for the experimental setups with predefined hyperparameter values, additional experimental results, prompts used in this paper, and the produced heuristics, respectively.

### 4.1   DERIVING PENALTY HEURISTICS FOR GLS TO SOLVE TSP

In this subsection, we exploit Hercules and Hercules-P to derive penalty heuristics for Guided Local Search (GLS) to solve the Travelling Salesman Problem (TSP). The seed function is human-designed heuristic KGLS (Arnold & Sörensen, 2019). We choose three LLM-based HG algorithms as benchmarking models, namely Random, EoH (Liu et al., 2024a), and ReEvo (Ye et al., 2024a). Random is

---

[1]https://anonymous.4open.science/r/ICLR-12808

Table 1: Performance comparison of different GLS algorithms on TSP

| Algorithm | Type | Gain (%) ($n = 100$) | Gain (%) ($n = 200$) |
|---|---|---|---|
| KGLS-Random | GLS+Llama3-70b | -137.13 | 0.47 |
| KGLS-EoH (ICML'24) | GLS+Llama3-70b | -369.10 | 5.82 |
| KGLS-ReEvo (NeurIPS'24) | GLS+Llama3-70b | -661.69 | 2.19 |
| KGLS-Hercules-P (ours) | GLS+Llama3-70b | -218.91 | 4.71 |
| KGLS-Hercules (ours) | GLS+Llama3-70b | -12.48 | 3.42 |
| KGLS-Random | GLS+GPT-4o-mini | 63.64 | 3.44 |
| KGLS-EoH (ICML'24) | GLS+GPT-4o-mini | 25.53 | 5.62 |
| KGLS-ReEvo (NeurIPS'24) | GLS+GPT-4o-mini | -280.79 | 2.45 |
| KGLS-Hercules-P (ours) | GLS+GPT-4o-mini | **71.05** | 7.46 |
| KGLS-Hercules (ours) | GLS+GPT-4o-mini | 42.98 | **11.10** |

Table 2: Search cost comparison of different LLM-based HG algorithms on TSP

| Algorithm | Gain (%) | Time (m) | Context Token (k) | Generation Token (k) | |
|---|---|---|---|---|---|
| KGLS-Random | $3.44_{\pm 1.20}$ | $28.5_{\pm 2.2}$ | **0.2** | **19.4** | GPT-4o-mini |
| KGLS-EoH (ICML'24) | $5.62_{\pm 1.83}$ | $37.2_{\pm 7.2}$ | 43.5 | 26.2 | |
| KGLS-ReEvo (NeurIPS'24) | $2.45_{\pm 10.93}$ | $37.7_{\pm 12.2}$ | 95.5 | 42.0 | |
| KGLS-Hercules-P (ours) | $7.46_{\pm 5.36}$ | $23.6_{\pm 3.0}$ | 143.4 | 31.2 | |
| KGLS-Hercules (ours) | $11.10_{\pm 0.69}$ | $30.6_{\pm 1.4}$ | 95.8 | 33.3 | |

a straightforward method that derives heuristics directly using LLMs without incorporating search directions and is commonly used as a baseline model in NAS studies (Li & Talwalkar, 2020). In addition, unless specified otherwise, for the performance of LLM-based HG algorithms, namely Random, EoH, ReEvo, Hercules-P, and Hercules, we report the average performance of three independent runs, following the prior study (Ye et al., 2024a). The average gains of the heuristics produced by these algorithms are presented in Table 1, where $n$ denotes the problem scale. The gain measure is calculated as 1-(the performance of the LLM-produced heuristics)/(the performance of the original KGLS). In addition, in Appendix E.1, the performance of these derived heuristics is compared with SOTA algorithms LKH3 (Helsgaun, 2017) and EAX (Nagata & Kobayashi, 2013).

As shown in Table 1, for the 200-node TSP, the heuristics produced by Hercules using GPT-4o-mini outperform those produced by the other HG algorithms, yielding the best performance gain of 11.1%. In addition, when GPT-4o-mini is adopted, the average gain of Hercules-P drops by only 3.64% comparing to Hercules, securing the second-best performance. EoH ranks at the third place in the gain metric. The experimental results shown in Table 1 highlight that the choice of LLM significantly impacts the performance of the produced heuristics. Nevertheless, Hercules and Hercules-P consistently outperform ReEvo across all node scales, regardless of the LLM in use.

Table 2 presents the search cost comparison of LLM-based HG algorithms across four metrics, namely gain (identical to the bottom-right cell of Table 1), search time, context token, and generation token. The results show that Hercules yields better gains without substantially increasing the costs of context and generation tokens, compared to ReEvo. Moreover, ReEvo and EoH spend longer search time when compared to the others, likely due to their ineffective search directions, which cause the LLM to derive complex but suboptimal heuristics. The std value of 10.93 for ReEvo further underscores this issue. On the other hand, Hercules-P reduces the overall search time to 77% (23.6/30.6) of that required by Hercules. Although Hercules-P uses approximately 1.5 times more context tokens than Hercules and ReEvo, it does not significantly increase the cost of generation tokens, which are typically more expensive (OpenAI). This makes Hercules-P ideal for environments with limited computing resources. Notably, Random utilizes only 0.2k context tokens, because of its simple prompts used for heuristic generation. However, this simplicity limits its ability to derive well-performing heuristics.

## 4.2 DERIVING CONSTRUCTIVE HEURISTICS TO SOLVE TSP

To assess the generalization capabilities of Hercules and Hercules-P across different HG tasks, we employ them in this subsection to derive constructive heuristics, which sequentially select unvisited nodes for solving real-world TSPLIB benchmarks (Reinelt, 1991). The seed function is genetic programming hyper-heuristic (Duflo et al., 2019). As shown in Table 3, Hercules achieves the highest average gain of 4.87% across eighteen TSPLIB instances, followed by EoH with the average gain of 4.8%. In contrast, both Random and ReEvo perform poorly, yielding negative gains on average, i.e., failing to improve the performance of the seed function.

Table 3: Performance comparison of different constructive heuristic algorithms on TSPLIB

| instances (total number) | Random | EoH (ICML'24) | ReEvo (NeurIPS'24) | Hercules-P (ours) | Hercules (ours) | |
|---|---|---|---|---|---|---|
| $n < 101$ (4) | -3.92 | **16.68** | 1.18 | 14.16 | 10.52 | GPT-3.5-turbo |
| $101 \leq n \leq 500$ (9) | -3.80 | -0.60 | -1.17 | 0.71 | **2.25** | |
| $n > 500$ (5) | -5.73 | **5.32** | 0.46 | 0.95 | 5.18 | |
| Avg. Gain (%) (18) | -4.49 | 4.80 | -0.16 | 3.42 | **4.87** | |

Table 4: Performance comparison of different ACO algorithms on BPP and MKP

| Algorithm | Type | BPP (Gain (%)), LLM: Llama3.1-405b | | | MKP (Gain (%)), LLM: Gemma2-27b | | |
|---|---|---|---|---|---|---|---|
| | | $n = 120$ | $n = 500$ | $n = 1,000$ | $n = 120$ | $n = 500$ | $n = 1,000$ |
| ACO+Random | ACO+LLM | $0.00_{\pm 0.00}$ | $-0.09_{\pm 0.04}$ | $0.00_{\pm 0.04}$ | $1.24_{\pm 0.03}$ | $3.21_{\pm 1.17}$ | $4.01_{\pm 1.59}$ |
| ACO+EoH (ICML'24) | ACO+LLM | $0.14_{\pm 0.12}$ | $0.16_{\pm 0.35}$ | $0.38_{\pm 0.53}$ | $\underline{1.61}_{\pm 0.48}$ | $4.42_{\pm 1.10}$ | $5.81_{\pm 1.40}$ |
| ACO+ReEvo (NeurIPS'24) | ACO+LLM | $\underline{0.66}_{\pm 0.50}$ | $\underline{1.49}_{\pm 0.25}$ | $2.01_{\pm 0.34}$ | $1.59_{\pm 0.72}$ | $4.67_{\pm 0.95}$ | $\underline{6.31}_{\pm 0.38}$ |
| ACO+Hercules-P (ours) | ACO+LLM | $0.08_{\pm 0.08}$ | $1.47_{\pm 0.16}$ | $\underline{2.04}_{\pm 0.16}$ | $1.44_{\pm 0.38}$ | $\underline{4.73}_{\pm 0.90}$ | $6.14_{\pm 1.21}$ |
| ACO+Hercules (ours) | ACO+LLM | $\mathbf{0.84}_{\pm 0.14}$ | $\mathbf{1.64}_{\pm 0.17}$ | $\mathbf{2.19}_{\pm 0.20}$ | $\mathbf{1.99}_{\pm 0.50}$ | $\mathbf{6.40}_{\pm 0.97}$ | $\mathbf{8.22}_{\pm 1.17}$ |

Table 5: Performance comparison of different NCO solvers on TSP and CVRP

| Algorithm | Type | TSP (Gain (%)) | | | CVRP (Gain (%)) | | |
|---|---|---|---|---|---|---|---|
| | | $n = 200$ | $n = 500$ | $n = 1,000$ | $n = 200$ | $n = 500$ | $n = 1,000$ |
| POMO+Random | NCO+GPT-4o-mini | **3.05** | -18.90 | -35.10 | **3.07** | 1.14 | **2.86** |
| POMO+EoH (ICML'24) | NCO+GPT-4o-mini | 2.19 | 1.42 | 1.47 | 0.48 | -1.83 | 0.27 |
| POMO+ReEvo (NeurIPS'24) | NCO+GPT-4o-mini | 2.38 | -5.24 | -2.78 | 0.34 | -14.20 | -3.01 |
| POMO+Hercules-p (ours) | NCO+GPT-4o-mini | -0.10 | -4.81 | -3.58 | -0.57 | -3.29 | -0.57 |
| POMO+Hercules (ours) | NCO+GPT-4o-mini | 2.49 | **6.62** | **16.43** | 1.53 | **1.22** | 1.59 |
| LEHD+Random | NCO+GPT-4o-mini | 9.93 | **8.83** | 5.44 | 1.72 | 2.33 | 1.68 |
| LEHD+EoH (ICML'24) | NCO+GPT-4o-mini | **10.67** | 7.73 | 6.09 | 6.62 | 3.57 | 0.47 |
| LEHD+ReEvo (NeurIPS'24) | NCO+GPT-4o-mini | 6.94 | -1.78 | 1.56 | 10.19 | 4.97 | 0.70 |
| LEHD+Hercules-p (ours) | NCO+GPT-4o-mini | 9.55 | 7.53 | **6.89** | 4.44 | 2.45 | 0.75 |
| LEHD+Hercules (ours) | NCO+GPT-4o-mini | 7.46 | 6.64 | 5.14 | **14.37** | **7.90** | **2.33** |

## 4.3 DERIVING HEURISTIC MEASURES FOR ACO TO SOLVE BPP AND MKP

In this subsection, we exploit Hercules and Hercules-P to derive heuristic measures for Ant Colony Optimization (ACO) applied to the Bin Packing Problem (BPP) and Multiple Knapsack Problem (MKP). The seed function is a conventional ACO algorithm (Dorigo et al., 2006). We adopt Llama3.1-405b to solve BPP while adopt Gemma2-27b to solve MKP. This is because Llama3.1-405b fails to improve the seed function of MKP regardless of which LLM-based HG algorithm is executed. As shown in Table 4, Hercules outperforms the other algorithms across all COPs and LLMs, with particularly strong performance observed when solving the 1,000-scale MKP, achieving an 8.22% gain. In addition, when using Llama3.1-405b, Random fails to derive superior heuristics compared to the original ACO, while EoH achieves only a modest improvement, falling short when compared to the more substantial gains obtained by ReEvo, Hercules-P, and Hercules. In Appendix E.2, we further assess the performance of Hercules under varying ACO hyper-parameters.

## 4.4 RESHAPING ATTENTION SCORES FOR NCO TO SOLVE TSP AND CVRP

Recently, Wang et al. (2024) demonstrated that reshaping attention scores can enhance the generalization performance of NCO solvers trained on small-scale COPs for solving large-scale COPs. To assess the effectiveness of Hercules and Hercules-P on NCO solvers, following (Ye et al., 2024a), we select DAR (Wang et al., 2024) as the seed function for TSP and the vanilla POMO (Kwon et al., 2020) and LEHD (Luo et al., 2023) as seed functions for Capacitated Vehicle Routing Problem (CVRP). As shown in Table 5, Random outperforms the other four LLM-based HG algorithms on certain tasks. A plausible reason for this is that the LLM corpora may lack sufficient knowledge of emerging NCO domains, thus limiting the performance of the other four LLM-based HG algorithms. Nevertheless, the heuristics derived by Hercules outperform the corresponding seed functions across a wider range of tasks compared to Random. For example, Hercules performs better than Random on the 500- and 1,000-node scales for the TSP-POMO task. In addition, Appendix E.3 presents additional results of these LLM-based HG algorithms, when the adopted LLM is GLM-4-0520. Finally, Appendix E.4 provides a detailed comparison on search time across these five LLM-based HG algorithms. The experimental results show that Hercules-P achieves the shortest search time across all NCO tasks. For example, it solves the 1,000-node CVRP-LEHD task in roughly five hours, which is

Table 6: Ablation study results on different design choices

| Algorithm | Gain (%) | Algorithm | Gain (%) | Algorithm | Gain (%) | Algorithm | Gain (%) | |
|---|---|---|---|---|---|---|---|---|
| w/o CAP | 3.12 | Hercules ($\lambda = 0.5$) | 5.96 | w/o ConS | -4.06 | Hercules-P ($\delta = 0.2$) | 7.01 | GPT-4o-mini |
| w/o rank-based selection | 8.49 | Hercules ($\lambda = 0.9$) | 8.90 | w/o EXEMPLAR | -0.30 | Hercules-P ($\delta = 0.3$) | 6.21 | |
| | | Hercules ($\lambda = 1$) | 5.60 | | | | | |
| Hercules (w/o PPP) | **11.10** | Hercules ($\lambda = 0.7$) | **11.10** | Hercules-P | **7.46** | Hercules-P ($\delta = 0.1$) | **7.46** | |

approximately 41% of the time needed by Hercules. Across all tasks, Hercules-P effectively reduces the search time by 7%~59% when compared to Hercules.

### 4.5 Ablation Studies

In this subsection, we conduct ablation studies to investigate the effectiveness of the design choices of Hercules and Hercules-P, and present the results in Table 6. The adopted HG task is deriving penalty heuristics for GLS to solve TSPs (see Section 4.1). Specifically, w/o CAP refers to the setting using RP to provide search directions, w/o rank-based selection refers to the setting that randomly selects parent heuristics, w/o ConS refers to the setting that PPP assumes all predictions are accurate, and w/o EXEMPLAR refers to the setting that heuristic examples are randomly selected from the current population. For all the other experiments presented in this paper, $\lambda = 0.7$ is applied for Hercules, and $\delta = 0.1$ is applied for Hercules-P. As shown in Table 6, when CAP is omitted, the gain decreases by 7.98%, further demonstrating that CAP produces more specific search directions. In addition, the proposed rank-based selection mechanism significantly contributes to the superior performance of Hercules. For Hercules-P, ConS effectively determines unreliable predictions, preventing them from negatively affecting the derivation of high-performance heuristics. Finally, when EXEMPLAR is omitted, the gain decreases by 7.76%, mainly due to the associated degradation in predictive accuracy (elaborated in the following paragraph).

We further present the predictive accuracy of PPP with and without EXEMPLAR, both of which are executed ten times, aiming to perform meaningful statistical tests. In addition, we include w/ EXEMPLAR-U as an additional setting, where EXEMPLAR is able to select heuristics with identical fitness values. To assess whether different versions of EXEMPLAR can accurately predict the fitness values of the produced heuristics, we need to set a quantifying measure. Specifically, we intuitively deem a prediction accurate if the absolute error between the predicted fitness value and the true fitness value is less than $\delta \cdot (ub_t - lb_t)$. As shown in Figure 5, the inclusion of EXEMPLAR improves the median of predictive accuracy by 26% and 37% (both significantly different: $p =$ 0.048 and 0.004) when compared to w/ EXEMPLAR-U and w/o EXEMPLAR, respectively. In addition, the Pearson correlation coefficient analysis reveals a correlation coefficient of 0.39, indicating a moderate linear relationship between the predicted and true values. The one-way ANOVA test results yield a $p$-value of 0.6, suggesting that the mean difference between the predicted and true values is not statistically significant. It is imperative to clarify that although the proposed PPP may seem less accurate in predicting heuristic performance, the values shown in Figure 5 are determined by a strict measure of fitness values as afore-defined and they do not exhibit a strong correlation with the overall performance of Hercules-P, because many produced heuristics are reevaluated (see ConS in Section 3.2). As discussed in Sections 4.1 and 4.4, Hercules-P reduces search time by 7%~59% when compared to Hercules, while achieving on-par gain. We strongly believe that PPP is highly beneficial for HG tasks that require rapid solutions, e.g., deriving heuristics for the dynamic, near-real-time allocation of resources in 5G mobile edge cloud networks (Laboni et al., 2024). We plan to extend PPP by integrating it with other methods, such as beam search, to further enhance its predictive accuracy.

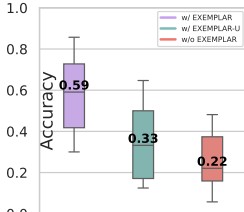

Figure 5: Ablation study on different EXEMPLAR variants.

### 5 Conclusion

To derive well-performing heuristics, we propose Hercules, which exploits our proprietary CAP to abstract the core components from elite heuristics, to produce more specific search directions. In addition, we introduce Hercules-P, a resource-efficient variant that integrates CAP with our novel PPP. PPP exploits previously evaluated heuristics to predict the performance of newly produced ones, thereby reducing the required computing resources for heuristic evaluations. The experimental results demonstrate the effectiveness of Hercules, Hercules-P, and all our designed mechanisms.

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

## A   DERIVATION OF INFORMATION GAIN FORMULA IN CAP

**Proposition 1.** *The information gain from abstracting core components is equal to:*

$$IG(\Omega_t) = -\sum_{j=0}^{k} p_j \log p_j \in (0, \log(k+1)]. \tag{8}$$

*Proof.*

$$
\begin{aligned}
IG(\Omega_t) &= H(\Omega_t) - p_0 H(\Omega_0) - \cdots - p_k H(\Omega_k) \\
&= -\sum_{i:\omega_i \in \Omega_t} p(\omega_i|\Omega_t) \log p(\omega_i|\Omega_t) \\
&\quad + p_0 \sum_{i:\omega_i \in \Omega_0} p(\omega_i|\Omega_0) \log p(\omega_i|\Omega_0) + \ldots \\
&\quad + p_k \sum_{i:\omega_i \in \Omega_k} p(\omega_i|\Omega_k) \log p(\omega_i|\Omega_k) \\
&= \sum_{i:\omega_i \in \Omega_0} p(\omega_i|\Omega_0) \left[ \log p(\omega_i|\Omega_0) - \log p(\omega_i|\Omega_t) \right] + \ldots \\
&\quad + \sum_{i:\omega_i \in \Omega_k} p(\omega_i|\Omega_k) \left[ \log p(\omega_i|\Omega_k) - \log p(\omega_i|\Omega_t) \right]
\end{aligned}
$$

According to the conditional probability, $p_j \cdot p(\omega_i|\Omega_j) = p(\omega_i|\Omega_t), \forall j \in \{0, 1, \cdots, k\}$. Thus, the $j$th term simplifies to the following expression:

$$
\begin{aligned}
&\sum_{i:\omega_i \in \Omega_j} p(\omega_i|\Omega_j) \left[ \log p(\omega_i|\Omega_j) - \log p(\omega_i|\Omega_t) \right] \\
&= \sum_{i:\omega_i \in \Omega_j} p(\omega_i|\Omega_j) \log \frac{p(\omega_i|\Omega_j)}{p(\omega_i|\Omega_t)} \\
&= -\sum_{i:\omega_i \in \Omega_j} p(\omega_i|\Omega_j) \log p_j \\
&= -p_j \log p_j
\end{aligned}
$$

Therefore, we conclude that:

$$IG(\Omega_t) = -\sum_{j=0}^{k} p_j \log p_j. \tag{9}$$

When $\forall j \in \{0, 1, \cdots, k\}, p_j = \frac{1}{k+1}$, $IG(\Omega_t)$ reaches its maximum value of $\log(k+1)$. When $\exists j \in \{0, 1, \cdots, k\}$ s.t. $p_j = 1$, $IG(\Omega_t)$ reaches its minimum value of 0. However, due to the diverse nature of LLM training corpora, the LLM will not consistently provide the same direction. Therefore, by abstracting core components, the unspecificity (entropy) can decrease within the $(0, \log(k+1)]$ interval. □

## B   THE SEARCH DIRECTIONS PRODUCED BY RP AND CAP

In this section, we present additional search directions produced by RP (Ye et al., 2024a) and CAP (our method) across various HG tasks, COPs and LLMs. Additionally, all produced unspecific search directions are highlighted in blue. For example, GPT-4o-mini frequently suggests the term "edge clustering", when performing RP. This direction "edge clustering" is frequently applied in tasks like recommendation systems, where it helps identify patterns in user interactions and preferences. However, it is not commonly used in heuristic algorithms for solving COPs and is, therefore, considered unspecific.

### Direction 1: The produced search directions for deriving penalty heuristics to solve TSP

```
# The LLM used to provide search directions is GPT-4o-mini.
RP:

Consider edge_clustering, incorporate historical_edge_frequencies, and adapt penalties dynamically based on
    current path exploration.

CAP:

Focus on relative edge scoring, incorporate multiple factors like connectivity and distance, and enhance
    normalization techniques.

# The LLM used to provide search directions is Llama-3-70b.
RP:

Normalize and symmetrize heuristics; consider the_opposite (not including an edge) for more effective
    penalties.

CAP:

Focus on relative edge costs (e.g., proximity concept) rather than absolute deviations from average distance.
```

### Direction 2: The produced search directions for deriving constructive heuristics to solve TSP

```
# The LLM used to provide search directions is GPT-3.5-turbo.
RP:

Use sorted future nodes by distance, limit look-ahead steps, and minimize total future score.

CAP:

Simplify heuristic by focusing on coefficients for better node selection efficiency.
```

### Direction 3: The produced search directions for deriving ACO heuristic measures to solve BPP

```
# The LLM used to provide search directions is Llama3.1-405b.
RP:

Consider non-linear relationships between demand ratios and heuristics, and experiment with different
    sparsification thresholds for better performance.

CAP:

Simplification and normalization of demand values can lead to more effective heuristics, reducing
    computational complexity.
```

### Direction 4: The produced search directions for deriving ACO heuristic measures to solve MKP

```
# The LLM used to provide search directions is Gemma2-27b.
RP:

Focus on item value density, consider weight imbalance across dimensions, and refine sparsity thresholds.

CAP:

Consider the interaction between dimensionality constraints and item value across dimensions.
```

### Direction 5: The produced search directions for reshaping attention scores of POMO to solve TSP

```
# The LLM used to provide search directions is GPT-4o-mini.
RP:

Incorporate dynamic adjustment of K based on node density.  Use heuristics from successful_TSP_solutions as
    weight modifiers. Explore edge_clustering to reduce focus on distant nodes.

CAP:

Prioritize distance quantiles, and apply exponential decay for promising edges while suppressing undesirable
    ones more effectively.
```

**Direction 6: The produced search directions for reshaping attention scores of POMO to solve CVRP**

```
# The LLM used to provide search directions is GPT-4o-mini.
RP:

Incorporate route_clustering, demand distribution analysis, and consider multi-vehicle interactions for
    enhanced heuristics.

CAP:

Emphasize vectorization over loops for performance. Enhance demand penalties to better reflect capacity
    constraints. Normalize distances effectively to balance contributions.
```

**Direction 7: The produced search directions for reshaping attention scores of LEHD to solve TSP**

```
# The LLM used to provide search directions is GPT-4o-mini.
RP:

Incorporate edge_connectivity to prioritize clusters. Consider spatial locality using coordinates for
    refinement. Adaptively adjust weights based on current_solution_state.

CAP:

Use logarithmic scaling for distances, increase top-K selection, and implement normalization for better
    convergence and stability.
```

**Direction 8: The produced search directions for reshaping attention scores of LEHD to solve CVRP**

```
# The LLM used to provide search directions is GPT-4o-mini.
# N=200
RP:

Utilize matrix operations for demand calculations to enhance efficiency. Introduce adaptive penalties based
    on demand-to-capacity ratios. Explore additional features, like clustering of nodes, to improve routing
    logic.

CAP:

Focus on vectorized operations, minimize nested loops, penalize exceeding capacity more effectively, and
    integrate distance-demand balancing.

# N=500

RP:

Consider integrating real-time clustering and demand_forecasting for optimized routing. Explore adaptive
    penalties and multi-objective criteria.

CAP:

Prioritize vectorized operations, minimize nested loops, reward feasible short connections, and enhance
    penalties for exceeding capacities.

# N=1,000
RP:

Incorporate vehicle_utilization_metrics. Explore clustering techniques. Include demand prioritization based
    on proximity. Optimize candidate edge selection dynamically.  Use adaptive penalties for infeasible
    edges. Consider adding multiple objectives in assessment.

CAP:

Incorporate vectorized calculations, normalize scores, and prioritize low-distance/high-demand paths for
    improved efficiency and effectiveness.
```

# C  THE ADOPTED CROSSOVER, ELITIST MUTATION OPERATORS, AND OTHER EC DEFINITIONS

For Hercules and Hercules-P, each heuristic code snippet denotes an individual within the population. Notably, these individuals are not restricted by any predefined encoding format, apart from complying with a specified function signature (see Appendix F). Parent heuristics refer to the heuristics selected according to 5. They are utilized during the crossover and mutation processes to derive offspring heuristics. Elite heuristics denote the top-$k$ heuristics selected based on corresponding fitness values within the current iteration. During population initialization, we employ a simple prompt proposed by Ye et al. (2024a) to guide the LLM in randomly deriving the initial population.

For consistency, we adopt the crossover and mutation operators from the prior study (Ye et al., 2024a) in all the experiments presented in this paper. Specifically, for the adopted crossover operator, two distinct parent heuristics are selected according to (5). Subsequently, the relative fitness values of these two heuristics determine which one serves as the primary learning exemplar for deriving an offspring heuristic. The employed mutation operator is elitist mutation, which derives multiple heuristics based on the historically best heuristic, aiming to produce high-performance ones. The prompting formats for both the crossover and elitist mutation operators, as well as the other promptings (e.g., CAP and PPP) used in this paper are shown in Appendix F.

## D   DETAILED HYPER-PARAMETERS AND EXPERIMENTAL SETUPS

**Hyper-parameters**   In Table 7, we present the hyper-parameters of Hercules and Hercules-P. In addition, following the prior study (Ye et al., 2024a), the temperature of the LLM is added by 0.3 to enhance the diversity of the initial population.

**Hardware**   We comprehensively evaluate the performance of all algorithms, using a computer equipped with an Intel(R) Xeon(R) W-2235 CPU.

Table 7: Parameters of Hercules and Hercules-P

| Parameter | Value |
| --- | --- |
| LLM temperature | 1 |
| Population size $N$ | 15 |
| CAP coefficients $k, \lambda$ | 5, 0.7 |
| Maximum number of evaluations | 100 |
| Crossover rate | 1 |
| Mutation rate | 0.5 |
| ConS coefficients $\delta, \alpha, \beta$ | 0.1, 0.5, 0.8 |

To ensure a fair comparison, we adopt the parameter configurations of all seed functions (e.g., KGLS parameters) as specified in the prior study (Ye et al., 2024a), which also documented the definitions of all HG tasks used in this paper. In addition, following the prior study (Ye et al., 2024a), the performance metric for TSP and CVRP is the gap, which is defined as the relative difference in the "average length" between corresponding heuristics and LKH3 (Helsgaun, 2017). For BPP and MKP, the performance metrics are the number of bins used and the total profit, respectively. Finally, for all experiments in this paper, we exploit the training and test datasets to derive well-performing heuristics and assess the final derived heuristics, respectively. Specifically, during the search process, the performance of heuristics on the training datasets determines their fitness values. The heuristic with the best performance on the training dataset is selected as the final derived heuristic. We then further assess the performance of all final derived heuristics on test datasets and report the experimental results in Section 4. In the following part of this section, we present the details of training datasets and test datasets of all HG tasks.

**Generating Penalty Heuristics for Guided Local Search**   During the search process, the performance of newly produced heuristics is evaluated using a training dataset comprising the number of 20 TSP instances, each with 200 nodes. Subsequently, we assess the performance of the final derived heuristics on two test datasets and report the results. Both test datasets contain 64 TSP instances, but differ in node scale, with one consisting of 100-node instances and the other of 200-node instances. All instances in both training and test datasets are uniformly distributed.

**Generating Constructive Heuristics**   During the search process, the performance of newly produced heuristics is evaluated on a training dataset comprising the number of 64 TSP instances, each with 50 nodes, following a uniform distribution. Subsequently, the performance of the final derived heuristics on TSPLIB instances is reported in Table 3.

**Generating Heuristic Measures for Ant Colony Optimization**   For BPP, during the search process, the performance of heuristics is evaluated on the training dataset consisting of 30 instances with 500 items each. The three test datasets each consist of 1,000 instances, with 120, 500, and

1,000 items, respectively. The bin capacity across all instances is fixed at 150, and item sizes are uniformly sampled from the range $[20, 100]$.

For MKP, the training dataset includes 30 instances, each with 120 items. The three test datasets each consist of 1,000 instances, with 120, 500, and 1,000 items, respectively. Both item values and weights are uniformly sampled from the range $[0, 1]$.

**Reshaping Attention Scores for Neural Combinatorial Optimization**  For TSP-POMO and CVRP-POMO tasks, during the search process, the performance of newly produced heuristics is evaluated on a training dataset comprising 64 instances, each with 200 nodes. Subsequently, we report the performance of the final derived heuristics on three test datasets of different scales, namely 200-node, 500-node, and 1,000-node scales. Each test dataset contains 64 instances. All instances are uniformly distributed. In addition, for CVRP-POMO, customer locations are uniformly sampled within the unit square, and customer demands are drawn from the discrete set $\{1, 2, \ldots, 9\}$, each vehicle's capacity is set to 50, and the depot is centrally located in the unit square.

For the TSP-LEHD task, during the search process, the performance of newly produced heuristics is evaluated on a training dataset consisting of 64 instances, each with 200 nodes. Subsequently, we report the performance of the final derived heuristics on three test datasets, namely 200-node, 500-node, and 1,000-node datasets, each containing 64 instances. Both the training and test datasets are sourced from (Luo et al., 2023). For the CVRP-LEHD task, following the prior study (Ye et al., 2024a), we apply LLM-based HG algorithms to derive heuristics for three training datasets, corresponding to problem sizes of $n = 200$, 500, and $1,000$, respectively. Subsequently, we assess these final derived heuristics on the corresponding scale test datasets and report the experimenatl results. The training dataset for $n = 200$ consists of 64 instances, while those for $n = 500$ and $n = 1,000$ contain 32 instances each. All test datasets consist of 64 instances. In addition, all the training and test datasets are sourced from (Luo et al., 2023).

# E  ADDITIONAL EXPERIMENT RESULTS

## E.1  COMPARISON OF THE DERIVED HEURISTICS AND SOTA ALGORITHMS

In this subsection, we present the gap for various algorithms, where gap denotes the relative difference in the "average length" between corresponding heuristics and LKH3 (Helsgaun, 2017). For these LLM-based HG algorithms, we report the average gap of heuristics derived from GPT-4o-mini. As shown in Table 8, Hercules outperforms EAX (Nagata & Kobayashi, 2013), achieving a gap of 0.237% relative to LKH3.

Table 8: Performance comparison of different heuristic algorithms on 200-node TSP

| Algorithm | Gap (%) |
|---|---|
| LKH3 (Helsgaun, 2017) | - |
| EAX (Nagata & Kobayashi, 2013) | 4.859 |
| KGLS (Arnold & Sörensen, 2019) | 0.267 |
| KGLS+Random | 0.258 |
| KGLS+EoH (ICML'24) | 0.251 |
| KGLS+ReEvo (NeurIPS'24) | 0.260 |
| KGLS+Hercules-P (ours) | 0.247 |
| KGLS+Hercules (ours) | **0.237** |

## E.2  ABLATION STUDY ON DIFFERENT ACO HYPER-PARAMETER

In this subsection, to further assess the robustness of Hercules under varying ACO hyper-parameters, we reduce the population size of ACO from 20 to 10. The adopted LLM is Llama3.1-405b. As shown in Table 9, the experimental results demonstrate that even under this more stringent condition, Hercules consistently outperforms Random, EoH, and ReEvo, achieving a gain of 0.93%. In addition, Table 9 includes the execution times of ACO and LLM-derived ACO variants. The experimental results indicate that LLM-derived ACO variants do not significantly increase execution time, compared with the original ACO.

Table 9: Ablation study results on different ACO hyper-parameter

| Algorithm | BPP ($n = 120$) | |
| --- | --- | --- |
| | Gain (%) | Time (s) |
| ACO | - | 261 |
| ACO+Random | -0.60 | 263 |
| ACO+EoH (ICML'24) | 0.25 | 264 |
| ACO+ReEvo (NeurIPS'24) | 0.20 | 268 |
| ACO+Hercules-P (ours) | 0.46 | 264 |
| ACO+Hercules (ours) | **0.59** | 267 |

### E.3 Additional Experiments of Reshaping Attention Scores for NCO

In this subsection, following the prior study (Ye et al., 2024a), we adopt GLM-4-0520 as LLM to further assess the performance of Hercules for solving large-scale TSP_LEHD task. In addition, it is important to emphasize that in the experiments conducted for this subsection, the fitness evaluation function during the search process is tailored to the problem size of the corresponding test dataset, ensuring consistency between the scales used for searching and testing. As shown in Table 10, Hercules achieves the best performance on datasets with 200 and 500 nodes, whereas Hercules-P outperforms on the 1,000-node scale, achieving a gain of 11.72% over the seed function.

Table 10: Performance comparison of different LLM-based HG algorithms on TSP_LEHD task

| Algorithm | Type | TSP (Gain (%)) | | |
| --- | --- | --- | --- | --- |
| | | $n = 200$ | $n = 500$ | $n = 1,000$ |
| LEHD+Random | NCO+GLM-4-0520 | 8.48 | 8.36 | 7.70 |
| LEHD+EoH (ICML'24) | NCO+GLM-4-0520 | 10.84 | 9.47 | 8.06 |
| LEHD+ReEvo (NeurIPS'24) | NCO+GLM-4-0520 | 10.13 | 8.70 | 6.97 |
| LEHD+Hercules-p (ours) | NCO+GLM-4-0520 | 9.98 | 8.80 | **11.72** |
| LEHD+Hercules (ours) | NCO+GLM-4-0520 | **11.06** | **9.24** | 8.16 |

### E.4 Search Time Comparison of Diverse LLM-based HG Algorithms

In Table 11, we present the search time of different LLM-based HG algorithms across diverse NCO tasks. As shown in Table 11, Hercules-P outperforms the other LLM-based HG algorithms in terms of search time, while Random ranks at the second place. On these NCO tasks, Hercules-P reduce the search time by 48%, 7%, 31%, 27%, 38%, and 59%, respectively, when compared to Hercules. This reduction in search time is especially significant for large-scale COPs, where search can extend to several hours. In these cases, incorporating PPP demonstrates highly effective in reducing the resource expenditure.

Table 11: Search time comparison of different LLM-based HG algorithms on diverse HG tasks

| | Algorithm Task | Random | EoH (ICML'24) | ReEvo (NeurIPS'24) | Hercules-P (ours) | Hercules (ours) |
| --- | --- | --- | --- | --- | --- | --- |
| Time (m) | TSP-POMO | 15.95 | 18.17 | 17.89 | **11.50** | 22.12 |
| | CVRP-POMO | 16.86 | 30.54 | 29.57 | **9.51** | 10.28 |
| | TSP-LEHD | 30.58 | 39.55 | 37.25 | **28.72** | 41.43 |
| | CVRP-LEHD ($n = 200$) | 45.73 | 67.27 | 61.58 | **31.20** | 42.80 |
| | CVRP-LEHD ($n = 500$) | 149.31 | 224.01 | 215.61 | **110.28** | 178.01 |
| | CVRP-LEHD ($n = 1,000$) | 639.83 | 854.25 | 854.71 | **310.98** | 757.67 |

## F Prompts Used in Hercules and Hercules-P

Prompts used for Hercules or Hercules-P can be categorized as problem-specific prompts and general prompts. This section provides a detailed overview of the used general prompts, while problem-specific prompts (including the heuristic description, COP description, seed function, and function signature) are documented in the prior study (Ye et al., 2024a).

Prompt 9: System prompt for elitist mutation and crossover operators.

```
You are an expert in the domain of optimization heuristics. Your task is to design heuristics that can
effectively solve optimization problems.
Your response outputs Python code and nothing else. Format your code as a Python code string:
"```python ... ```".
```

Prompt 10: System prompt for abstracting core components.

```
You are an expert in the domain of automatic heuristics algorithm design. Your task is to give some hints for
Large Language Model evolutionary framework to evolve better heuristic methods.
```

Prompt 11: System prompt for providing search directions.

```
You are an expert in the domain of optimization heuristics. Your task is to give hints to design better
heuristics.
```

Prompt 12: System prompt for predicting heuristic performance.

```
You are an expert in the domain of heuristics evaluation. Your task is to predict the performance of
heuristics.
```

Prompt 13: User prompt for population initialization.

```
{task_description}

{seed_function}

Refer to the format of a trivial design above. Be very creative and give `{func_name}_v2`. Output code only and
enclose your code with Python code block: ```python ... ```.
```

Prompt 14: User prompt for abstracting core components.

```
The {func_name} function is a part of {alg} for solving {pro}.
{func_desc}

Below are five {func_name} functions:
[code_0]
{code_0}
-------
[code_1]
{code_1}
-------
[code_2]
{code_2}
-------
[code_3]
{code_3}
-------
[code_4]
{code_4}

Summarize the key code components of these functions that potentially influence the effectiveness and
performance of the algorithm, using less than 200 words.
```

Prompt 15: User prompt for providing short-term search directions.

```
Below are two {func_name} functions for {problem_desc}
{func_desc}
```

```
You are produced with two code versions below, where the second version performs better than the first one.

[Worse code]
{worse_code}

[Better code]
{better_code}

Below are some core components of the previous {func_name} functions.

[component]
{component}

Reflect about why the second code performs better than the first, considering the core components.
Only output some hints on designing better {func_name} functions base your reflections, using less than
20 words.
```

Prompt 16: User prompt for providing long-term search directions.

```
Below is your prior long-term search directions on designing heuristics for {problem_desc}
{prior_direction}

Below are some newly gained insights.
{new_direction}

Below are some core components of the previous {func_name} functions.

[component]
{component}

Write constructive hints for designing better heuristics, based on prior search directions, new insights, and
the core components, using less than 50 words.
```

Prompt 17: User prompt for crossover.

```
{task_description}

[Worse code]
{function_signature0}
{worse_code}

[Better code]
{function_signature1}
{better_code}

[direction]
{short_term_direction}

[Improved code]
Please write an improved function '{function_name}_v2', according to the search directions. Output code only
and enclose your code with Python code block: '''python ... '''.
```

Prompt 18: User prompt for elitist mutation.

```
{task_description}

[Prior direction]
{long-term_direction}

[Code]
{function_signature1}
{elitist_code}

[Improved code]
Please write a mutated function '{function_name}_v2', according to the search directions. Output code only
and enclose your code with Python code block: '''python ... '''.
```

Prompt 19: User prompt for predicting heuristic performance.

```
The {func_name} function is a part of {alg}, which is used to solve {pro}.
{func_desc}
Here are some example codes and their corresponding performance scores that you can refer to for
predicting heuristic functions:
```

```
[example code 0]
{code_0}
[performance score of example code 0]
{score_0}
---
[example code 1]
{code_1}
[performance score of example code 1]
{score_1}
---
[example code 2]
{code_2}
[performance score of example code 2]
{score_2}
---
[example code 3]
{code_3}
[performance score of example code 3]
{score_3}
---
[example code 4]
{code_4}
[performance score of example code 4]
{score_4}
---
[example code 5]
{code_5}
[performance score of example code 5]
{score_5}
---
[example code 6]
{code_6}
[performance score of example code 6]
{score_6}
---
[example code 7]
{code_7}
[performance score of example code 7]
{score_7}
---
[example code 8]
{code_8}
[performance score of example code 8]
{score_8}
---
[example code 9]
{code_9}
[performance score of example code 9]
{score_9}
---
Here are some codes that you need to predict:
[code_10]
{code_10}
---
[code_11]
{code_11}
---
[code_12]
{code_12}
---
[code_13]
{code_13}
---
[code_14]
{code_14}
---
[code_15]
{code_15}
---
[code_16]
{code_16}
---
[code_17]
{code_17}
---
[code_18]
{code_18}
---
[code_19]
{code_19}
Predict the performance of the above codes by comparing their semantic meanings with the produced example
codes. Provide a performance score and a confidence number based on your evaluation for each code. The
performance score should be a float within the range [{score_0}, {score_1}], where a lower score indicates a
better-performing heuristic. The confidence number should be a float within the range [0,1], indicating how
similar the semantics of the code is to the most similar example code. Note that you can only give a confidence
level = 1 if the code is semantically identical to the produced example code. Output only the performance score
and confidence number of these codes that need to be predicted, strictly adhering to the following format. No
other words and punctuation should be included in the output.
'''code_10: score, confidence,
code_11: score, confidence,
code_12: score, confidence,
code_13: score, confidence,
```

```
code_14: score, confidence,
code_15: score, confidence,
code_16: score, confidence,
code_17: score, confidence,
code_18: score, confidence,
code_19: score, confidence'''
```

# G    LLM-DERIVED HEURISTICS

## G.1    HEURISTICS PRODUCED BY EoH

In this subsection, we present three final EoH-derived heuristics using Llama3.1-405b for solving BPP. It can be seen that, when Llama3.1-405b is is adopted, EoH cannot derive intricate heuristics, which is why it performs poorly in solving BPP.

EoH 1: The ACO heuristic measure produced by Hercules using Llama3.1-405b for solving BPP.

```python
def EoH_1(demand: np.ndarray, capacity: int) -> np.ndarray:
    demand_ratio = demand / capacity
    return np.tile(np.power(demand_ratio, 2), (demand.shape[0], 1)) * (1 - demand_ratio[:, np.newaxis])

def EoH_2(demand: np.ndarray, capacity: int) -> np.ndarray:
    demand_ratio = demand / capacity
    return np.tile(demand_ratio, (demand.shape[0], 1)) * (1 - demand_ratio[:, np.newaxis])

def EoH_3(demand: np.ndarray, capacity: int) -> np.ndarray:
    residual_capacity = capacity - demand[:, None]
    return (demand[None, :] <= residual_capacity) / (1 + np.abs(residual_capacity - demand[None, :]))
```

## G.2    HIGH-PERFORMANCE HEURISTICS PRODUCED BY HERCULES

In this subsection, we present the best heuristics produced by Hercules for all tasks.

Heuristic 1: The high-performance GLS heuristic produced by Hercules using GPT-4o-mini for solving TSP.

```python
def heuristic(distance_matrix: np.ndarray) -> np.ndarray:
    n = distance_matrix.shape[0]
    heuristics_scores = np.zeros_like(distance_matrix)

    # Penalty function for edges based on both distance and connectivity
    for i in range(n):
        for j in range(n):
            if i != j:
                # Increase penalty for longer distances
                heuristics_scores[i, j] = distance_matrix[i, j]

                # Reduce score if this edge is part of a densely connected horizon
                connections = np.sum(distance_matrix[i] < distance_matrix[i, j]) - 1 # excluding itself
                heuristics_scores[i, j] *= (1 + connections * 0.1) # penalizing connected edges more

    return heuristics_scores
```

Heuristic 2: The high-performance constructive heuristic produced by Hercules using GPT-3.5-turbo for solving TSP.

```python
def heuristic(current_node: int, destination_node: int, unvisited_nodes: set, distance_matrix: np.ndarray) -> int:
    """Select the next node to visit from the unvisited nodes with look-ahead mechanism."""

    def calculate_look_ahead_score(node, next_node):
        if len(unvisited_nodes) == 1:
            return 0
        lookahead_nodes = unvisited_nodes - {next_node}
        min_lookahead_distance = min([distance_matrix[node][i] for i in lookahead_nodes if i != node])
        return -0.1 * min_lookahead_distance

    c1, c2, c3, c4 = 0.4, 0.3, 0.2, 0.1
    scores = {}
    for node in unvisited_nodes:
        all_distances = [distance_matrix[node][i] for i in unvisited_nodes if i != node]
        average_distance_to_unvisited = np.mean(all_distances)
        std_dev_distance_to_unvisited = np.std(all_distances)
        lookahead_score = calculate_look_ahead_score(current_node, node)
        score = c1 * distance_matrix[current_node][node] - c2 * average_distance_to_unvisited + c3 * \
            std_dev_distance_to_unvisited - c4 * distance_matrix[destination_node][node] + lookahead_score
        scores[node] = score
    next_node = min(scores, key=scores.get)
    return next_node
```

Heuristic 3: The high-performance ACO heuristic measure produced by Hercules using Llama3.1-405b for solving BPP.

```python
def heuristic(demand: np.ndarray, capacity: int) -> np.ndarray:
    """
    This function calculates the heuristics for the Bin Packing Problem (BPP).
    Parameters:
    demand (np.ndarray): A 1D array representing the sizes of the items.
    capacity (int): The capacity of each bin.
    Returns:
    np.ndarray: A 2D array where heuristics[i][j] represents how promising it is to put item i and item j in
        the same bin.
    """
    # Calculate the complementarity of each pair of items
    # The complementarity is the difference between the capacity and the sum of the demands of the two items
    complementarity = capacity - np.add.outer(demand, demand)

    # Apply exponential decay to the complementarity values
    # This reduces the dominance of large values and emphasizes the importance of small values
    decayed_complementarity = np.exp(-complementarity / capacity)

    # Normalize the demand values to be between 0 and 1
    normalized_demand = demand / demand.max()

    # Calculate the heuristic value for each pair of items
    # The heuristic value is the product of the normalized demands and the decayed complementarity
    heuristics = np.outer(normalized_demand, normalized_demand) * decayed_complementarity

    # Sparsify the matrix by setting unpromising elements to zero
    # Here, we consider elements with a value less than 0.5 as unpromising
    heuristics[heuristics < 0.5] = 0

    return heuristics
```

Heuristic 4: The high-performance ACO heuristic measure produced by Hercules using Gemma2-27b for solving MKP.

```python
def heuristic(prize: np.ndarray, weight: np.ndarray) -> np.ndarray:
    prize_per_unit_weight = prize / np.sum(weight, axis=1)
    max_weight_ratios = np.max(weight / np.expand_dims(np.sum(weight, axis=1), axis=1), axis=1)
    density_score = prize_per_unit_weight * (1 - max_weight_ratios)

    # Weight Magnitude Awareness
    weight_magnitude = np.sum(weight, axis=1)
    magnitude_bonus = np.exp(-weight_magnitude / np.max(weight_magnitude))

    # Distribution Awareness with Adaptive IQR
    density_percentile_75 = np.percentile(density_score, 75)
    density_percentile_25 = np.percentile(density_score, 25)
    iqr = density_percentile_75 - density_percentile_25
    adaptive_iqr_window = 0.3 * iqr
    distribution_factor = np.where(density_score > density_percentile_75, 1.2,
            np.where(density_score > density_percentile_75 - adaptive_iqr_window, 1, 0.5))

    # Dimensionality-Weighted Density Scores (Tighter Coupling and Exponent Tuning)
    dimensionality_weights = np.sum(weight > 0, axis=1) / weight.shape[1]
    dimensionality_bonus = density_score ** (1 + dimensionality_weights * 2)

    # Sparsity Penalty
    sparsity_penalty = np.where(np.sum(weight > 0, axis=1) < weight.shape[1] , 1.2, 1)

    heuristics = density_score * magnitude_bonus * distribution_factor * dimensionality_bonus *
        sparsity_penalty
    heuristics[heuristics < np.percentile(heuristics, 5)] = 0

    return heuristics
```

Heuristic 5: The high-performance POMO heuristic produced by Hercules using GPT-4o-mini for solving TSP.

```python
def heuristic(distance_matrix: torch.Tensor) -> torch.Tensor:
    """
    heuristics computes a refined heuristic for TSP based on the distance matrix by evaluating edges
    and applying adaptive, non-linear transformations for better edge prioritization.
    The heuristic incorporates clustering dynamics and balances exploration-exploitation strategies.
    """
    distance_matrix[distance_matrix == 0] = 1e5
    K = 5 # Top-K nearest neighbors for refined edge selection
    alpha = 0.9 # Increased weight for promoting close edges
    beta = 0.1 # Reduced weighting factor for penalizing distant edges
    epsilon = 1e-5 # Small constant to prevent division by zero

    # Start with heuristic values based on a transformation of the distance matrix
    heu = -distance_matrix.clone()

    # Find the top-K nearest neighbors
    _, indices = torch.topk(distance_matrix, k=K, largest=False, dim=1)

    # Create masks for top-K edges
```

```
1350    topk_mask = torch.zeros_like(distance_matrix, dtype=torch.bool)
1351    topk_mask.scatter_(1, indices, True)
1352
        # Adaptive transformations on selected edges with logarithmic weighting
1353    transformation_term = -alpha * torch.log(1 + distance_matrix[topk_mask])
        penalty_term = beta * (1 / (distance_matrix[topk_mask] + epsilon))
1354
        # Combine results for top-K and retain default penalties elsewhere
1355    heu[topk_mask] = transformation_term + penalty_term
1356
        # Employ edge clustering insights by grouping nearly equal distances
1357    distance_mean = distance_matrix.mean(dim=1, keepdim=True)
        distance_std = distance_matrix.std(dim=1, keepdim=True)
1358    cluster_mask = torch.abs(distance_matrix - distance_mean) < distance_std
1359
        # Apply a refinement for edges within the same cluster with increased adjustment
1360    heu[cluster_mask] += 0.3 # Increased favor for edges within the same cluster
1361
        # Additional adjustment for edges based on their proximity to the mean distance
1362    solution_proximity = distance_matrix.mean() # Example proximity metric
        adjustment_term = heu - (distance_matrix - solution_proximity)
1363    heu += adjustment_term * 0.15 # Slightly refine penalties based on distance to the mean solution proximity
1364    return heu
1365
```

Heuristic 6: The high-performance POMO heuristic produced by Hercules using GPT-4o-mini for solving CVRP.

```
1368    def heuristic(distance_matrix: torch.Tensor, demands: torch.Tensor) -> torch.Tensor:
        """Enhanced adaptive heuristic function for CVRP with refined scoring aggregation and weight parameters."""
1369
        # Total vehicle capacity, normalized to the highest demand
1370    vehicle_capacity = demands.max()
1371
        # Initialize distance scores (negative for minimization)
1372    distance_scores = -distance_matrix.clone()
1373
        # Compute combined demand interactions with broadcasting
1374    demand_matrix = demands.unsqueeze(1) + demands.unsqueeze(0) # Shape (n, n)
1375
        # Identify edges exceeding vehicle capacity
        exceeding_capacity_mask = demand_matrix > vehicle_capacity
1376
        # Calculate demand scores with adaptive penalties and strong incentives for valid demands
1377    demand_scores = torch.where(
          exceeding_capacity_mask,
1378      -5 * (demand_matrix - vehicle_capacity) ** 2, # Higher penalty for exceeding capacity
1379      3 * (vehicle_capacity - demand_matrix)  # Incentive for satisfying demands
        )
1380
        # Combine distance and demand scores with an aggregation weight
1381    alpha = 0.7 # Weight for distance scoring
        beta = 0.3 # Weight for demand scoring
1382    combined_scores = alpha * distance_scores + beta * demand_scores
1383
        # Normalize combined scores for consistent indicator range
1384    combined_scores_normalized = (combined_scores - combined_scores.min()) / (combined_scores.max() -
1385        combined_scores.min() + 1e-10)
1386    return combined_scores_normalized
1387
```

Heuristic 7: The high-performance LEHD heuristic produced by Hercules using GPT-4o-mini for solving TSP.

```
1390    def heuristic(distance_matrix: torch.Tensor) -> torch.Tensor:
        """
1391    Improved heuristics for the TSP utilizing adaptive thresholds, robust statistical measures,
        and dynamic edge scoring systems to enhance edge desirability evaluation.
1392    """
        distance_matrix[distance_matrix == 0] = 1e5
1393    N = distance_matrix.size(0)
1394
        # Calculate mean and robust median as a central tendency measure
1395    mean_distances = distance_matrix.mean(dim=1, keepdim=True)
        median_distances = distance_matrix.median(dim=1, keepdim=True).values
1396
        # Calculate edge scores based on how far they deviate from both mean and median
1397    deviations_from_mean = -(distance_matrix - mean_distances) / (mean_distances + 1e-5)
        deviations_from_median = -(distance_matrix - median_distances) / (median_distances + 1e-5)
1398
        # Initialize heuristic scores with a combination of deviations
1399    heuristics_scores = (deviations_from_mean + deviations_from_median) / 2
1400
        # Apply dynamic proximity boosts for edges that are closer than a weighted threshold
1401    dynamic_threshold = 0.5 * (mean_distances + median_distances)
        proximity_boosts = torch.where(distance_matrix <= dynamic_threshold,
1402                      (1 / N * dynamic_threshold - distance_matrix).clamp(min=0),
1403                      torch.tensor(0.0, device=distance_matrix.device))
```

```
1404        # Update heuristic scores with proximity boosts
1405        heuristics_scores += proximity_boosts
1406
            return heuristics_scores
1407
```

## Heuristic 8: The high-performance LEHD heuristic produced by Hercules using GPT-4o-mini for solving CVRP.

```
1410    #N=200
1411    def heuristic(distance_matrix: torch.Tensor, demands: torch.Tensor) -> torch.Tensor:
1412        """An improved heuristic implementation for the Capacitated Vehicle Routing Problem (CVRP) with refined
                dynamic penalties and transformations."""
1413
1414        vehicle_capacity = 1.0 # Normalize demands with respect to maximum capacity
            num_customers = demands.shape[0]
1415
1416        # Create a matrix for combined demand
            demand_matrix = demands.unsqueeze(1) + demands.unsqueeze(0) # Shape: [n, n]
1417
            # Create a mask for viable connections based on vehicle capacity
1418        is_viable = (demand_matrix <= vehicle_capacity).float()
1419        # Compute distance scores, avoiding self-distances by adding a large penalty
            distance_scores = 1 / (distance_matrix + torch.eye(num_customers) * 1e6)
1420
1421        # Calculate promising indicators
            promising_indicators = is_viable * distance_scores
1422
1423        # Dynamic penalties based on excess demand
            excess_demand_penalty = (demand_matrix - vehicle_capacity).clamp(min=0)
            penalty_factor = excess_demand_penalty ** 2 / (vehicle_capacity ** 2 + 1e-6)
1424        promising_indicators -= penalty_factor * (distance_scores * 2 - 1)
1425        # Clustering for improved route planning with a more responsive threshold
1426        cluster_threshold = 0.3 # Adaptive threshold for clustering based on distance
            clusters = (distance_matrix < cluster_threshold).float()
1427        promising_indicators *= clusters
1428        # Normalize scores to range between -1 and 1
1429        min_value = promising_indicators.min()
            max_value = promising_indicators.max()
1430
1431        if max_value != min_value:
                promising_indicators = (promising_indicators - min_value) / (max_value - min_value) * 2 - 1
1432        # Enhance promising connections via a non-linear transformation
1433        promising_indicators = promising_indicators ** 3 * torch.sign(promising_indicators + 1e-6) # Added epsilon
                for stability
1434
1435        return promising_indicators
1436    #N=500
        def heuristic(distance_matrix: torch.Tensor, demands: torch.Tensor) -> torch.Tensor:
1437        """Enhanced heuristic implementation for Capacitated Vehicle Routing Problem that evaluates edge
                desirability."""
1438        num_customers = demands.shape[0]
            vehicle_capacity = 1.0 # Normalized capacity
1439
1440        # Initialize cost matrix
            cost_matrix = distance_matrix.clone()
1441
1442        # Calculate total demand and initialize demand density
            demand_density = demands / demands.sum()
1443        total_demand_matrix = demands.unsqueeze(1) + demands.unsqueeze(0)
1444        # Calibrated penalties for demand violation
            penalties = (total_demand_matrix > vehicle_capacity).float() * 3.0 # Increased penalties for more emphasis
1445
            # Evaluate edge desirability based on demand compatibility and distance
1446        mask_compatible = total_demand_matrix <= vehicle_capacity
1447        mask_incompatible = total_demand_matrix > vehicle_capacity
1448        # Adjust cost matrix based on compatibility and added penalties
            cost_matrix[1:, 1:] = torch.where(mask_compatible[1:, 1:], -distance_matrix[1:, 1:], distance_matrix[1:,
1449            1:] * penalties[1:, 1:])
1450        # For depot connections, favorably adjust edges
1451        cost_matrix[0, 1:] = -distance_matrix[0, 1:] * 0.5 # Strongly favor depot-to-customer
            cost_matrix[1:, 0] = -distance_matrix[1:, 0] * 0.5 # Strongly favor customer-to-depot
1452
            # Return normalized desirability
1453        return cost_matrix
1454
        #N=1,000
1455    def heuristic(distance_matrix: torch.Tensor, demands: torch.Tensor) -> torch.Tensor:
1456        n = distance_matrix.shape[0]
            vehicle_capacity = 1.0 # normalized vehicle capacity
1457        heuristic_scores = torch.zeros_like(distance_matrix)

            # Create a mask for valid edges based on capacity constraints (non-self-loops)
```

```
demand_within_capacity = (demands.unsqueeze(1) + demands.unsqueeze(0) <= vehicle_capacity) & (
    distance_matrix != 0)

# Calculate effective distance score
effective_distances = torch.where(distance_matrix > 0, 1.0 / (distance_matrix + 1e-6), torch.zeros_like(
    distance_matrix))

# Initialize promising edges
heuristic_scores[demand_within_capacity] = effective_distances[demand_within_capacity]

# Assign stronger penalties for infeasible edges
heuristic_scores[~demand_within_capacity] = -200.0 # Strong penalty for infeasible edges

# Scale scores for promising paths using min-max normalization
positive_scores = heuristic_scores[heuristic_scores > 0]

if positive_scores.numel() > 0:
    min_positive = positive_scores.min()
    max_positive = positive_scores.max()

    # Normalize to [0, 1]
    heuristic_scores[heuristic_scores > 0] = (heuristic_scores[heuristic_scores > 0] - min_positive) / (
        max_positive - min_positive)

# Apply additional penalties based on demand
demand_excess = demands.unsqueeze(1) - vehicle_capacity
demand_excess[demand_excess < 0] = 0 # No penalty for nodes within capacity
heuristic_scores -= demand_excess * 15.0 # Apply strong penalty for edges leading to high demand

return heuristic_scores
```

