# OpenReview forum: "Efficient Heuristics Generation for Solving Combinatorial Optimization Problems Using Large Language Models"
_ICLR.cc/2025/Conference — Submitted to ICLR 2025_

### Official Review · Reviewer_f4Hd · 2024-10-28

**Soundness:** 2
**Presentation:** 3
**Contribution:** 2
**Rating:** 5
**Confidence:** 3

**Summary:**

This paper explores the application of LLM in autonomously generating heuristics to solve COPs and proposes a novel algorithm named Hercules to address the two main challenges of existing approaches.

Hercules utilizes the Core Abstraction Prompting (CAP) method to abstract core components from elite heuristics and incorporate them as prior knowledge in prompts, thereby reducing the specificity of search directions. This paper further introduces Hercules-P, an efficient variant of Hercules that integrates CAP with the novel Performance Prediction Prompting (PPP) method. PPP leverages LLMs to predict the fitness values of newly derived heuristics based on their semantic similarity to previously evaluated ones, significantly reducing the required computing resources.

**Strengths:**

1. The paper is laudable for its well-structured and logical presentation, providing a comprehensive understanding of the research topic.

2. The article is praiseworthy for its extensive experimental data and significant findings. The authors have selected a number of baselines for comparative experiments on different benchmarks.

3. The supplement provided in this article is adequate. It explains in detail for the reader what is not expanded in detail in the paper, including specific experimental data, hyperparameter settings, Critical Difference Analysis, etc.

**Weaknesses:**

1. In the literature of TSP and CVRP, it is known that those conventional heuristic algorithms, such as LKH [1] and EAX [2], exhibit robust performance. It appears, however, that this submission does not address LKH and EAX, nor does it provide a comparative analysis of the proposed algorithm against these established methods.

1. In lines 85-100 of the Introduction section, the authors describe two challenges to LLM-based HG methods, mentioning in the second challenge that these methods introduce numerous linear operations and conditional branches that make the GPU less efficient for these algorithms. In lines 113-128, the authors claim to have proposed Hercules-P in order to better address the second challenge, but I don't seem to have read in the manuscript how Hercules-P reduces linear operations and conditional branches, making GPUs more efficient in processing these algorithms. May I ask if the authors have solved this challenge? If not, these representations are inappropriate.

2. Does the appearance of CVRP in Section 4.4 stand for Capacitated Vehicle Routing Problem? The authors do not explain what CVRP stands for in the body of the manuscript, and the only explanation appears in the code comments in the Appendix section (line 1337). This cannot be very clear to the reader when it comes to understanding the manuscript.

3. In Section 2.3, the authors mention two challenges for NCO solvers: improving generalisation capabilities and large-scale COPs performance. In Table 5, for LEHD, the performance improvement of either Hercules or Hercules-p gradually decreases as the problem size of TSP or VCRP increases. Does this mean that Hercules also fails to address the challenges faced by NCO solvers? Can Hercules still provide performance gains when the problem size is larger? Further discussion is requested from the authors.


## References
[1] Keld Helsgaun. General k-opt submoves for the Lin-Kernighan TSP heuristic. Mathematical Programming Computation 1(2-3): 119-163 (2009)

[2] Yuichi Nagata, Shigenobu Kobayashi. A Powerful Genetic Algorithm Using Edge Assembly Crossover for the Traveling Salesman Problem. INFORMS Journal on Computing 25(2): 346-363 (2013)

**Questions:**

Please reply to my comments in "Weaknesses".

**Details Of Ethics Concerns:**

This submission does not have ethics concerns.

---

### Official Review · Reviewer_QjjZ · 2024-11-01

**Soundness:** 3
**Presentation:** 3
**Contribution:** 3
**Rating:** 8
**Confidence:** 4

**Summary:**

The paper proposes a framework to use LLMs to generate heuristics for solving
optimization problems. The authors describe their framework and evaluate it
empirically, comparing to other approaches in the literature.

**Strengths:**

The proposed framework is interesting and seems to work well in practice.

**Weaknesses:**

The choice of KGLS as seed heuristics should be justified as it was not designed
for the general TSP. Why not LKH? This should also be considered in the
empirical evaluation; in particular to answer the question of whether KGLS is a
reasonable heuristic to start with in this case (improving over a weak heuristic
is easier than improving over a strong heuristic).

Figure 5 has no axis labels.

**Questions:**

The differences are small in some cases and it would be great if the authors could
provide error bounds or confidence intervals for the empirical results.

Why is KGLS is reasonable base heuristic?

Update after responses: Thank you for your responses. I have updated my score.

---

### Official Review · Reviewer_KGdN · 2024-11-04

**Soundness:** 2
**Presentation:** 2
**Contribution:** 1
**Rating:** 3
**Confidence:** 4

**Summary:**

This paper studies the generation of heuristics for combinatorial optimization problems using LLMs.

The work continues similar work in this space that tries to mimic evolutionary computation (crossover, mutation) via LLMs. The result is an LLM-infused metaheuristic algorithm.

Unlike the previous work, the paper claims 1) to address introducing more problem specificity into the prompts and 2) to speed up the process by using LLMs to predict the performance of generated accuracies to skip over evaluating them fully.

Overall, I enjoyed reading this paper, and I appreciated the work that went into building an end-to-end pipeline with several components.

**Strengths:**

Integrating LLMs with heuristic solving is an exciting combination.
The paper implements an end-to-end pipeline that starts with a seed query that then mimics evolutionary computing via LLMs, yielding heuristics that can be embedded in the Local Search Meta-Heuristics for different combinatorial problems.
The connection with Information Gain is an excellent addition
From a practical perspective, the paper considers several details into account such as reducing costs via LLM predictors.

**Weaknesses:**

As rightly noted in the paper, the idea of mimicking evolutionary computation via LLMs is not new. In fact, most (all?) crossover and mutation operators are from Ye et. al. 2024. On the one hand, the experiments and the ablation study show that the proposed modifications might offer some benefit in the results, and on the other hand, they can be regarded as incremental, and it is not clear what's the main takeaway.

Regarding the presentation, I found it difficult/confusing that many moving parts are introduced as large components with several acronyms Hercules, CAP, PPP, EXEMPLAR, Cons --but after all, the provided pseudocode shows the overall algorithm, so I am not sure what these abstractions add to the presentation. Also, the paper claims our "propriety" CAP algorithm  --what does that mean?

The idea of adding more specificity to the prompts seems reasonable at a high level, but the paper overindexes too much into the example in Figure 1. The information gain analysis is interesting (and is borrowed from a Hu et. al. 2024) but at the end what happens is we select top-k core components. And that's also not uniform, we do that only some number of iterations (denoted by \lambda in the paper), all of which remain as more hyper-parameters to deal with.

The experiments cover TSP, CVPR, Binpacking, Multi-Knapsacks. Importantly, the starting seed function seems critical to the approach. The method generates heuristics but the overall approach to solve these problems are meta-heuristics.  (please correct me if I understand this correctly). For TSP, we use guided local search. For BinPacking and Knapsacks we use Ant-Colony Optimization. One might argue that the settings of the outer meta-heuristics and their performance are crucial to the overall results and not just the heuristics (generated by LLMs here.) The experiments do not discuss or study any of this.

Additionally, all comparisons are with other LLM-based heuristics generations. Note that this is quite a costly approach (hence some effort with performance predictors to save time etc.). According to the tables in the appendix, we are consuming many many minutes upto 5 hours. Then, it is not clear to me how to fairly evaluate these results. How does the same GLS and ACO without the advanced heuristics found by LLM but with standard heuristics perform given the same amount of time? (Btw, does this time include LLM queries or only running the heuristics after the LLM generates them against the instances?)

This might not be surprising that the choice of LLM quite affects the results (Table 1; LLama vs GPT-4o). But then it makes one wonder how much of the value comes from the many moving components proposed here vs. plain and simple, the underlying LLM.

**Questions:**

Could you provide details on the outer meta-heuristics (GLS, ACO etc.)? How much of the results are due to the LLM integration with CAP, PPP etc. vs. the meta-heuristics leading the search into good solutions.
It would be interesting to know the comparison between default GLS, ACO or even other baselines for TSP, BinPacking, MKP to position the results in this paper. As is, it is hard to evaluate the significance

---

> ### Author Response · Authors · 2024-11-25
> **Response to Reviewer KGdN (Part 1)**
>
> Thank you for your appraisal. Here are our detailed responses to your comments. If anything remains unclear, we would be more than happy to provide further clarification.
>
> ---
> > **W1:** As rightly noted in the paper, the idea of mimicking evolutionary computation via LLMs is not new. In fact, most (all?) crossover and mutation operators are from Ye et. al. 2024. On the one hand, the experiments and the ablation study show that the proposed modifications might offer some benefit in the results, and on the other hand, they can be regarded as incremental, and it is not clear what's the main takeaway.
>
> **Response 2.1 (Doubt on contribution & takeaway):**
>
> Indeed, most evolutionary computation algorithms rely on crossover and mutation operators. The reason we adopt the crossover and mutation operators introduced by Ye et al. (2024a) is **to ensure a fair demonstration of the effectiveness of our proposed CAP.** As shown in Figure 1 of the previously submitted manuscript, **compared with RP proposed by Ye et al. (2024a), our proprietary CAP enhances the quality of the produced search directions by first prompting the LLMs to abstract the core components as prior knowledge.** Extensive experimental results further validate the advancement of CAP over RP, even when identical crossover and mutation operators are employed. **Additionally, we introduce a novel LLM-based heuristic performance predictor, named PPP, to mitigate the excessively long search times required by ReEvo (Ye et al., 2024a) for certain HG tasks.** Furthermore, PPP incorporates two tailored mechanisms, EXEMPLAR and Cons, to enhance predictive accuracy and identify unreliable predictions, respectively. **These contributions in our work (i.e., CAP, PPP, EXEMPLAR, and Cons) are unique, significantly enhancing the quality of the derived heuristics and reducing unnecessary resource expenditure compared to ReEvo. Therefore, we argue that our contributions should not be regarded as "incremental".**
>
> **We believe the main takeaways of our work had been clearly stated throughout.** For example, in the Abstract of the previously submitted manuscript, we highlighted:
>
> > "*To enable LLMs to provide specific search directions, we propose the Hercules algorithm, which leverages our designed Core Abstraction Prompting (CAP) method to abstract the core components from elite heuristics and incorporate them as prior knowledge in prompts*",
>
> and
>
> > "*To reduce the required computing resources for evaluating the derived heuristics, we propose few-shot Performance Prediction Prompting (PPP), a first-of-its-kind method for the Heuristic Generation (HG) task. PPP leverages LLMs to predict the fitness values of newly derived heuristics by analyzing their semantic similarity to previously evaluated ones. We further develop two tailored mechanisms for PPP to enhance predictive accuracy and determine unreliable predictions, respectively.*"
>
> Furthermore, in Section 1 of the previously submitted manuscript, we had summarized our contributions comprehensively. **In summary, our work centers around our proprietary CAP and PPP methods, which are the key contributions of this work.**
> > **W2:** Regarding the presentation, I found it difficult/confusing that many moving parts are introduced as large components with several acronyms Hercules, CAP, PPP, EXEMPLAR, Cons --but after all, the provided pseudocode shows the overall algorithm, so I am not sure what these abstractions add to the presentation. Also, the paper claims our "propriety" CAP algorithm --what does that mean?
>
> **Response 2.2 (Excessive acronyms & meaning of proprietary):**
>
> **We respectfully disagree with your observation that the mentioned acronyms make the presentation confusing. Instead, we properly defined all the acronyms of our proposed components in the previously submitted manuscript.** The use of acronyms is commonly adopted in most research publications to better facilitate the reference to the respective proposed models or components. For example, Liu et al. (2024a) defined five prompt strategies (E1, E2, M1, M2, and M3) to enhance readability and highlight their contributions. We are more than happy to provide additional explanations or engage in further discussion to clarify any points and ensure there is no misunderstanding.
>
> It should be a misunderstanding of yours that we never used the phrase "*our **propriety** CAP algorithm*". While, the term **proprietary** in "*our **proprietary** CAP algorithm*" indicates that CAP is a novel method proposed by us and, to the best of our knowledge, the first-of-its-kind to better address the issue of unspecificity in LLM responses within the field of LLM-based HG.
>
> Fei Liu, Xialiang Tong, Mingxuan Yuan, et. al., Evolution of heuristics: Towards efficient automatic algorithm design using large language
> mode. In ICML, 2024a.

---

> > ### Author Response · Authors · 2024-11-25
> > **Response to Reviewer KGdN (Part 2)**
> >
> > > **W3:** The idea of adding more specificity to the prompts seems reasonable at a high level, but the paper overindexes too much into the example in Figure 1. The information gain analysis is interesting (and is borrowed from a Hu et. al. 2024) but at the end what happens is we select top-k core components. And that's also not uniform, we do that only some number of iterations (denoted by \lambda in the paper), all of which remain as more hyper-parameters to deal with.
> >
> > **Response 2.3 (Significance of Figure 1 & purpose of $\lambda$):**
> >
> > Figure 1 shows a representative example to further explain the challenge faced by RP (i.e., unspecificity in LLM responses) and our motivation (i.e., enhancing the quality of the produced search directions by incorporating core components as prior knowledge), aiming to help readers better understand the distinction between our proposed CAP method and the RP method (Ye et al., 2024a). While Figure 1 is cited five times throughout the main text, most references (e.g., on Page 4) serve to explain the challenge faced by RP (three times). We deem these cross-references are necessary to enhance reader comprehension, and they are not overly indexed.
> >
> > Although the concept of analyzing information gain is inspired by Hu et al. (2024), **our work introduces extensive extensions**. Specifically, Hu et al. (2024) did not define the value range of $IG(\Omega_t)$, while we had rigorously proven that $IG(\Omega_t)$ can decrease through core component abstraction and eventually fall within the $(0,\log(k+1)]$ range. Furthermore, whereas Hu et al. (2024) restricted $\Omega_t$ to two subsets (i.e., "yes" and "no" subsets), we expanded it to accommodate $k+1$ subsets, broadening its applicability. For example, in Appendix A of the previously submitted manuscript, we stated:
> >
> > > "*Therefore, by abstracting core components, the unspecificity (entropy) can decrease within the $(0,\log(k+1)]$ interval.*"
> >
> > In addition, we select the top-$k$ core components based on their fitness values, **which is a standard practice in evolutionary computation and aligns with the principles of elitism** (Zhang et al., 2015).
> >
> > Regarding your concern about hyperparameter $\lambda$, we would like to emphasize that λ is primarily introduced to better balance between exploitation and exploration, as mentioned on Page 6 of the previously submitted manuscript:
> >
> > > “*In addition, Hercules adopts the core components of the top-$k$ heuristics as prior knowledge during the first $\lambda$ percent of iterations ($\lambda \in$ [0,1]). In the later iterations, to better preserve population diversity, Hercules directly applies the core components of the parent heuristics as prior knowledge to provide search directions, bypassing the abstraction process of elite heuristics*”.
> >
> > In addition, we would like to further clarify that except for ablation studies, all parameters (including $\lambda$) are set consistently across all experiments, as detailed on Page 18 of the previously submitted manuscript. Finally, it is worth noting that most evolutionary computation methods, e.g., (Zhan et al., 2009), (Yang et al., 2018), and (Zhang et al., 2021), utilize adaptive hyperparameters to effectively balance between exploitation and exploration. **Therefore, the incorporation of the hyperparameter $\lambda$ is consistent with established practices in the field of evolutionary computation.**
> >
> > Jianming Zhang, Weifeng Pan, Jingjing Wu, and Jing Wang. Top-$k$ elites based oppositional differential evolution. IJWMC, 2015.
> >
> > Zhi-Hui Zhan, Jun Zhang, Yun Li, and Henry Shu-Hung Chung. Adaptive particle swarm optimization. IEEE TCYB, 2009.
> >
> > Qiang Yang, Wei-Neng Chen, Jeremiah Da Deng, Yun Li, Tianlong Gu, and Jun Zhang. A level-based learning swarm optimizer for large-scale optimization. IEEE TEVC, 2018.
> >
> > Fangfang Zhang, Yi Mei, Su Nguyen, and Mengjie Zhang. Correlation coefficient-based recombinative guidance for genetic programming hyperheuristics in dynamic flexible job shop scheduling.
> > IEEE TEVC, 2021.

---

> > > ### Author Response · Authors · 2024-11-25
> > > **Response to Reviewer KGdN (Part 3)**
> > >
> > > > **W4:** The experiments cover TSP, CVPR, Binpacking, Multi-Knapsacks. Importantly, the starting seed function seems critical to the approach. The method generates heuristics but the overall approach to solve these problems are meta-heuristics. (please correct me if I understand this correctly). For TSP, we use guided local search. For BinPacking and Knapsacks we use Ant-Colony Optimization. One might argue that the settings of the outer meta-heuristics and their performance are crucial to the overall results and not just the heuristics (generated by LLMs here.) The experiments do not discuss or study any of this.
> > >
> > > **Response 2.4 (Choices of meta-heuristics):**
> > >
> > > We agree with you that the settings and performance of the outer meta-heuristics are crucial to the overall results. Therefore, we adopt most HG tasks introduced in the closely relevant prior study (Ye et al., 2024a), e.g., deriving penalty heuristics for GLS to solve TSP and deriving heuristic measures for ACO to solve BPP and MKP. In addition, the parameter configurations for all seed functions are set identically to those in the prior study (Ye et al., 2024a), as stated in Appendix D of the previously submitted manuscript:
> > >
> > > > “*To ensure a fair comparison, we adopt the parameter configurations of all seed functions (e.g., KGLS parameters) as specified in the prior study (Ye et al., 2024a)*”.
> > >
> > > **Nonetheless, we believe the primary focus of our work is not to compare the performance of GLS and ACO on TSP, but rather to find out how much the improvements may the Hercules-derived heuristics lead to when comparing against those derived by other LLM-based HG algorithms, all using these established seed functions.** We believe that our choices and parameter configurations of seed functions provide a robust and fair demonstration of Hercules’ advantages over EOH (ICML’24) and ReEvo (NeurIPS’24), which are the most relevant and state-of-the-art (SOTA) baselines regarding our work.
> > >
> > > To further address your concern, we have conducted an ablation study on the parameters of ACO to comprehensively evaluate Hercules’ robustness. The following experimental results clearly demonstrate that Hercules consistently outperforms the other LLM-based HG algorithms.
> > >
> > > | Algorithm | Gain (%) | Time (s) |
> > > | --- | --- | --- |
> > > | ACO | -   | 261 |
> > > | ACO+Random | -0.60 | 263 |
> > > | ACO+EoH (ICML'24) | 0.25 | 264 |
> > > | ACO+ReEvo (NeurIPS'24) | 0.20 | 268 |
> > > | ACO+Hercules-P (ours) | _0.46_ | 264 |
> > > | ACO+Hercules (ours) | **0.59** | 267 |
> > >
> > > > **W5:** Additionally, all comparisons are with other LLM-based heuristics generations. Note that this is quite a costly approach (hence some effort with performance predictors to save time etc.). According to the tables in the appendix, we are consuming many many minutes upto 5 hours. Then, it is not clear to me how to fairly evaluate these results. How does the same GLS and ACO without the advanced heuristics found by LLM but with standard heuristics perform given the same amount of time? (Btw, does this time include LLM queries or only running the heuristics after the LLM generates them against the instances?)
> > >
> > > **Response 2.5 (Performance of meta-heuristics):**
> > >
> > > As mentioned on Page 8 of the previously submitted manuscript:
> > >
> > > > “*The gain measure is calculated as 1-(the performance of the LLM-produced heuristics)/(the performance of the original KGLS)*.”
> > >
> > > This metric ensures that our experimental results always compare the performance of the LLM-derived heuristics against the original seed function. Additionally, as repeatedly stated in the previously submitted manuscript, the reported time refers to the overall search time of the LLM-based HG algorithms, including both the LLM response time and the execution time of all derived heuristics. For example, on Page 6 of the previously submitted manuscript, we stated:
> > >
> > > > "*Hercules-P reduces the overall search time to 77% (23.6/30.6) of that required by Hercules.*”
> > >
> > > It is generally recognized that LLM-derived heuristics do not significantly increase the execution time of the seed function. In response to your comment, we have now included execution time experiments in Appendix E.2 of the revised manuscript, comparing the original ACO with the final LLM-derived ACO variants. The experimental results (see the table presented in Response 2.4) demonstrate that the LLM-derived heuristics significantly improve the performance of the original ACO algorithm while only taking a marginally longer execution time.

---

> > > > ### Author Response · Authors · 2024-11-25
> > > > **Response to Reviewer KGdN (Part 4)**
> > > >
> > > > > **W6:** This might not be surprising that the choice of LLM quite affects the results (Table 1; Llama vs GPT-4o). But then it makes one wonder how much of the value comes from the many moving components proposed here vs. plain and simple, the underlying LLM.
> > > >
> > > > **Response 2.6 (Ablation studies & underlying LLM):**
> > > >
> > > > We do agree with you that the choice of LLM greatly affects the results. That is exactly why we conducted various experiments (see Tables 1, 3, and 4) to demonstrate that **no matter which LLM is in use, Hercules consistently outperforms the baseline models**. This success can be attributed to the contributions of each proposed component, with Hercules-P achieving comparatively strong results for the same reason. **As presented in Section 4.5 of the previously submitted manuscript, we had conducted extensive ablation studies to demonstrate the effectiveness of CAP, PPP, and the other tailored mechanisms.** These experimental results clearly demonstrate the value-adds of all proposed methods and mechanisms.
> > > >
> > > > Furthermore, we believe the underlying LLM you refer to closely resembles the **Random** algorithm compared in our manuscript. As mentioned on Page 8 of the previously submitted manuscript:
> > > >
> > > > > “*Random is a straightforward method that derives heuristics directly using LLMs without incorporating search directions and is commonly used as a baseline model in NAS studies (Li&Talwalkar, 2020).*”
> > > >
> > > > Therefore, we did make a direct comparison with the underlying LLM (i.e., the Random algorithm), and the experimental results showed that Hercules outperforms the underlying LLM.
> > > >
> > > > > **Q1:** Could you provide details on the outer meta-heuristics (GLS, ACO etc.)?
> > > >
> > > > **Response 2.7 (Details of outer meta-heuristics):**
> > > >
> > > > The rationale behind selecting specific seed functions for different COPs is elaborated in Response 2.4. For specific choices of GLS and ACO, we generally follow the closely relevant prior study (Ye et al., 2024a). If you would like to know more about the parameter configurations and algorithm descriptions of all selected seed functions, you may refer to Appendices C and D of the prior study (Ye et al., 2024a) for more details. This experimental design had been clearly explained in Appendix D of the previously submitted manuscript:
> > > >
> > > > > “*To ensure a fair comparison, we adopt the parameter configurations of all seed functions (e.g., KGLS parameters) as specified in the prior study (Ye et al., 2024a), which also documented the definitions of all HG tasks used in this paper.*”
> > > >
> > > >
> > > >
> > > > > **Q2:** How much of the results are due to the LLM integration with CAP, PPP etc. vs. the meta-heuristics leading the search into good solutions. It would be interesting to know the comparison between default GLS, ACO or even other baselines for TSP, BinPacking, MKP to position the results in this paper. As is, it is hard to evaluate the significance
> > > >
> > > > **Response 2.8 (Ablation studies & performance of meta-heuristics):**
> > > >
> > > > For a detailed discussion on the effectiveness of CAP and PPP, please refer to Response 2.6. Additionally, the rationale of not directly comparing the performance of different seed functions, such as GLS and ACO, is explained in Response 2.4. In summary, we believe our work explicitly and comprehensively demonstrates the contributions of CAP, PPP, EXEMPLAR, and ConS, and presents the performance of the meta-heuristics.

---

> > > > > ### Comment · Reviewer_KGdN · 2024-11-26
> > > > >
> > > > > Thank you for the detailed response!

---

> > > > > > ### Author Response · Authors · 2024-11-27
> > > > > >
> > > > > > Thank you for acknowledging that you have reviewed our responses to all your comments. We sincerely hope we have adequately addressed your concerns regarding our work. Please do let us know if you have any additional concerns, questions, or suggestions. We are more than happy to engage in further discussions to improve our research. We greatly appreciate your understanding and support.

---

> > > > > > > ### Author Response · Authors · 2024-12-02
> > > > > > >
> > > > > > > Dear Reviewer KGdN,
> > > > > > >
> > > > > > > Thank you once again for your insightful comments and helpful suggestions. As the author-reviewer discussion will end soon (< 24 hours from now), we would greatly appreciate it if you could take a moment to review our rebuttal. We sincerely hope our efforts and improvements are taken into consideration. Please let us know if you have any further questions or concerns.
> > > > > > >
> > > > > > > Best regards,
> > > > > > > Authors

---

### Official Review · Reviewer_jJ2Y · 2024-11-04

**Soundness:** 2
**Presentation:** 2
**Contribution:** 3
**Rating:** 5
**Confidence:** 3

**Summary:**

The paper presents Hercules, an LLM-based algorithm for generating heuristics for combinatorial problems. The paper seems to extend the framework in Ye et al., 2024 with a more advanced direction generation (based on identifying core components in heuristics) as well as an LLM-based fitness calculation. The experiments show gains over the baselines.

**Strengths:**

Strengths:
- The topic is interesting and of recent interest
- The approach (CAP and PPP) seems novel.
- The experiments show significant gains over the baselines in deriving penalty heuristics for guided local search, as well as more moderate gains on constructive heuristics for TSP, heuristic measures for ant colony optimization, and reshaping of attention scores in neural combinatorial optimization,

**Weaknesses:**

Weaknesses:
- I found the claim about information gain to be quite confusing.
	- First, a lot of information is missing: why the number of core components corresponds to the number of heuristics (can we not have multiple core components per heuristic or the same core component in multiple heuristics)? why do we assume that the set of all possible directions can be partitioned into mutually exclusive subsets that correspond to components (can we not have the same direction for multiple core components)?
	- Second, it is really not clear why the information gain means we get better heuristics (as indicated in lines 284-285)? If the components generated are of low-quality the directions may be of lower quality as well.

- Experimental evaluation:
	- It is not clear what is being reported under gain: the definition is based on "the performance of ..." but it is not clear how performance is measured.

- Writing: the writing could improve as a lot of information is not clearly presented. For example there are no clear definitions for a range of terms like parent heuristics, elite heuristics, etc.

- The paper does not provide significant insight into the impact of the proposed techniques (CAP and PPP) beyond the experimental results. For example, it would be interesting to show an analysis of the correlation between predicted fitness values and quality of heuristics.

**Questions:**

I would appreciate the authors response and clarification on the points listed under "weaknesses"

---

### Meta-Review · Area_Chair_kw1y · 2024-12-20

**Metareview:**

The paper presents prompt and other refinements operating on the top of an LLM to guide a meta-heuristic approach dedicated to combinatorial optimization.
The approach continues the work done by Fei Liu et al, 2024 (ICML) and Haoran Ye et al, 2024 (NeurIPS).
The improvements come from the proposed core abstraction prompting (CAP) and the performance prediction prompting (PPP). Reminiscent of NAS, the authors also provide a generation of heuristics samples (EXAMPLAR) and a confidence stratification module (ConS) enhancing PPP.

**Additional Comments On Reviewer Discussion:**

The many ingredients in the approach were diversely appreciated by the reviewers, despite the authors' rebuttals.

The area chair encourages the authors to revise the paper, notably clarifying the claims (CAP being proprietary while the approach is made publicly available), and hopes to see this revised version soon.

---

### Decision · Program_Chairs · 2025-01-22

Reject